# Improved in situ characterization of protein complex dynamics at scale with thermal proximity co-aggregation

Siyuan Sun[1,3], Zhenxiang Zheng[1,3], Jun Wang[1], Fengming Li[1], An He[1], Kunjia Lai[1], Shuang Zhang[1,2], Jia-Hong Lu[2], Ruijun Tian[1] & Chris Soon Heng Tan[1] ✉

Cellular activities are carried out vastly by protein complexes but large repertoire of protein complexes remains functionally uncharacterized which necessitate new strategies to delineate their roles in various cellular processes and diseases. Thermal proximity co-aggregation (TPCA) is readily deployable to characterize protein complex dynamics in situ and at scale. We develop a version termed Slim-TPCA that uses fewer temperatures increasing throughputs by over 3X, with new scoring metrics and statistical evaluation that result in minimal compromise in coverage and detect more relevant complexes. Less samples are needed, batch effects are minimized while statistical evaluation cost is reduced by two orders of magnitude. We applied Slim-TPCA to profile K562 cells under different duration of glucose deprivation. More protein complexes are found dissociated, in accordance with the expected down-regulation of most cellular activities, that include 55S ribosome and respiratory complexes in mitochondria revealing the utility of TPCA to study protein complexes in organelles. Protein complexes in protein transport and degradation are found increasingly assembled unveiling their involvement in metabolic reprogramming during glucose deprivation. In summary, Slim-TPCA is an efficient strategy for characterization of protein complexes at scale across cellular conditions, and is available as Python package at https://pypi.org/project/Slim-TPCA/.

Proteins are molecular workforces performing virtually all biological processes in cells but they seldom act alone. Instead, many are part of higher-order molecular machines commonly termed protein complexes that are assembled when needed from protein–protein interactions (PPIs)[1]. The interactions among proteins are often dynamic and regulated by various mechanisms including temporal post-translational modifications and spatial redistribution of proteins that changes protein functionalities[2]. The study of PPIs can uncover the underlying machinery of various cellular processes and reveal how their dysregulation could lead to diseases, thus providing insight for the development of new therapeutic treatments including the identification of new drug targets[3]. As such, identifying and characterizing the dynamics of PPIs and protein complexes are among the endeavors of biologists with increasing attention and research efforts in recent years[4–6].

Experimental methods for studying PPIs can be broadly classified into traditional hypothesis-driven methods (e.g., immunoprecipitation, yeast two-hybrid) and high-throughput methods. Many of the latter are coupled with protein mass spectrometry (MS) for protein identification which includes co-fractionation/elution and crosslinking

[1]Department of Chemistry and Research Center for Chemical Biology and Omics Analysis, College of Science, Southern University of Science and Technology, Shenzhen, Guangdong, China. [2]State Key Laboratory of Quality Research in Chinese Medicine, Institute of Chinese Medical Sciences, University of Macau, Zhuhai, Macau SAR, China. [3]These authors contributed equally: Siyuan Sun, Zhenxiang Zheng. ✉e-mail: christan@sustech.edu.cn

of protein complexes[7,8]. In addition, a variety of computational models have also been proposed for predicting PPIs[9]. While experimental methods have their own limitations, they are often hailed as the gold standard used in the validation of computational approaches. Traditional experimental methods such as Y2H[10], LUMIER[11] and AP-MS[4] can be scaled up to obtain large sets of PPI data but often suffer from higher false-negative rate that necessitated a combination of complementary experimental methods to furnish a more comprehensive interactome. However, these methods require either protein engineering (e.g., epitope tagging) or the use of antibodies specific to target proteins, and are not applicable in situ[12]. Importantly, current large-scale interactome profiling is carried out with specific cell lines under basal conditions, thus the conservation and dynamics of identified PPIs and protein complexes across cell lines, cellular states and physiological conditions are unclear.

In 2018, we conceived the thermal proximity co-aggregation (TPCA) method, pioneering the use of thermal proteome profiling to study the dynamics of PPIs[13] and protein complexes in situ and at scale. The underlying experimental method of TPCA profiling is similar to the TPP/MS-CETSA method where proteins are subjected to a gradient of denaturing temperature. The solubility of protein decreases with increasing temperature which is quantified with MS[14,15], resulting in a "melting curve" that is often influenced by protein-ligand interactions. TPP/MS-CETSA permits the measurement of protein-ligand binding with endogenous proteins[16,17], and is widely used in drug target deconvolution[18] and off-target studies[19] in intact cells. Our previous work revealed that signature of PPIs is embedded in thermal proteomics data[13] as similar thermal melting curves of co-aggregating or co-precipitating proteins in the same protein complex. Arguably, TPCA profiling is one of the few methods that provide an approach to detect PPIs *en masse* occurring in intact cellular environment[8,20].

The TPCA method for analyzing protein–protein interactions has been independently demonstrated and applied since its conception. For example, the principle of TPCA was applied to study the modulation of protein complexes across different phases of cell cycle[21,22]. The complexes that are regulated in different phases of cell cycle were shown to have high correlation with the corresponding biological activities. Hashimoto et al. further applied TPCA to profile the dynamics of host protein complexes in different stages of viral infection and demonstrated that the TPCA can be used to study PPIs between proteins from different species[23]. Moreover, TPCA had been demonstrated in thermal proteomic data obtained from tissue and blood samples[24] where subunits of protein complexes are grouped hierarchically based on their median melting temperature ($T_m$) to assess their assembly state across different tissues. In 2021, Kalxdorf et al. used the TPCA method to analyze complex formation during T-cell activation using changes in the distance between attachment proteins and the core of complexes[25]. To facilitate the use of TPCA for profiling PPIs, the computational workflow for TPCA analysis has been integrated into rTPCA[26] and ProSAP[27] software analysis packages.

Current TPCA workflow provides statistical evidence for the modulation of protein complexes using existing databases of protein complexes as references[28,29]. In recent years, the repertoire of protein complexes have expanded greatly with large scale interactome profiling projects[4-6] and the development of system-level complex profiling techniques[20] including co-fractionation-MS[30-32] and protein network-based clustering algorithms[33,34]. However, the conservation and functions of these putative complexes across different cell types and cellular states are unclear.

Conceptually, without the need for antibodies and epitope-tagging of proteins, the TPCA method can be readily deployed to rapidly profile the dynamics of multiple uncharacterized protein complexes simultaneously under different cellular conditions and perturbations. This could aid in the function annotation of these complexes. Nonetheless, there are several aspects where the TPCA method can be further optimized. In the original TPCA method, 10 samples are heated to different temperatures in gradient followed by abundance quantification of soluble protein using protein MS. The difference in thermal solubility between proteins is measured by Euclidean distance, and the significance is estimated by a bootstrap sampling algorithm[13]. Thus, a large number of samples and long computational time is needed in conventional TPCA method. Furthermore, one set of isobaric tandem mass tags (TMT) reagents with 10 channels is used to label only one set of TPCA experiment for a condition, thus the control and experimental samples had to be analyzed by MS at different times with batch effect that contributes to deviation in measurements[35].

Here, we optimize the TPCA method and propose a streamlined version called Slim-TPCA. Slim-TPCA was optimized in both algorithmic and experimental aspects. First, in Slim-TPCA protocol, fewer temperature points are used which provide more flexibility in experimental design and help reduce batch effect (Fig. 1a). Similar ideas have been shown to be used to improve the TPP/MS-CETSA method. For example, the PISA[36] and iTSA[37] methods analyze different treatment conditions within the same TMT set for target deconvolution[19,38]. Next, through algorithm optimization, Slim-TPCA can better identify PPIs and the dynamic changes in protein complexes with improved statistical power. We applied the optimized Slim-TPCA method to analyze the dynamic changes of protein complexes at five different time points of glucose deprivation on MS concurrently. We successfully identified many protein complexes known to associate with glucose starvation and revealed connections with many other complexes, of which identified modulated Emerin complex 1 and USP22-SAGA complexes are experimentally validated by co-IP. Slim TPCA is available as an independent python package to facilitate analysis of protein complexes.

## Results
### Fewer temperature points in Slim-TPCA
The fundamental basis of TPCA profiling for characterizing protein–protein interactions and their dynamics is the similarity in their melting curves quantified as solubility similarity of interacting proteins across multiple denaturing temperatures. In the first version of TPCA method, samples are subjected to 10 gradient temperatures, followed by TMT labeling and MS analysis to infer the so-called "melting" curve of proteins. Such a workflow requires a large number of samples, entailing multiple sets of TMT reagents to label and analyze samples from different conditions (e.g., drug group and control group with replicates), which can lead to batch variation in MS measurement. As such, we seek to modify the original TPCA approach. Here, we assess whether fewer temperatures can be used to analyze PPIs with minimal loss in information. We first assess the feasibility of doing so using thermal solubility data of K562 generated from cell lysate and intact cell that were published in the proof-of-concept paper[13] (Supplementary Data 1–2). The data is used to evaluate the TPCA signature between proteins for differentiating known PPIs from random protein pairs, and the predictive power is quantified with Area Under the Curve (AUC) of the Receiver Operating Characteristic (ROC) curve. In a nutshell, high AUC value indicates higher predictive (differentiating) power. Here, TPCA signature is used qualitatively to refer clustering of data points that possibly arise from thermally-induced proximity co-aggregation of proteins.

First, we analyzed the predictive power of different number of temperature points by testing all combination of temperatures with Euclidean distance measure used in the original work (Fig. 1b). As expected, the AUC of ROC decreases with fewer temperature points but the decrease is surprisingly gradual. For example, the median AUC of two, three and four temperature points (including mandatory 37 °C) is 0.65, 0.68 and 0.69 compared to AUC of 0.71 with 10 temperature points. Thus, we conclude that TPCA analysis can be performed using

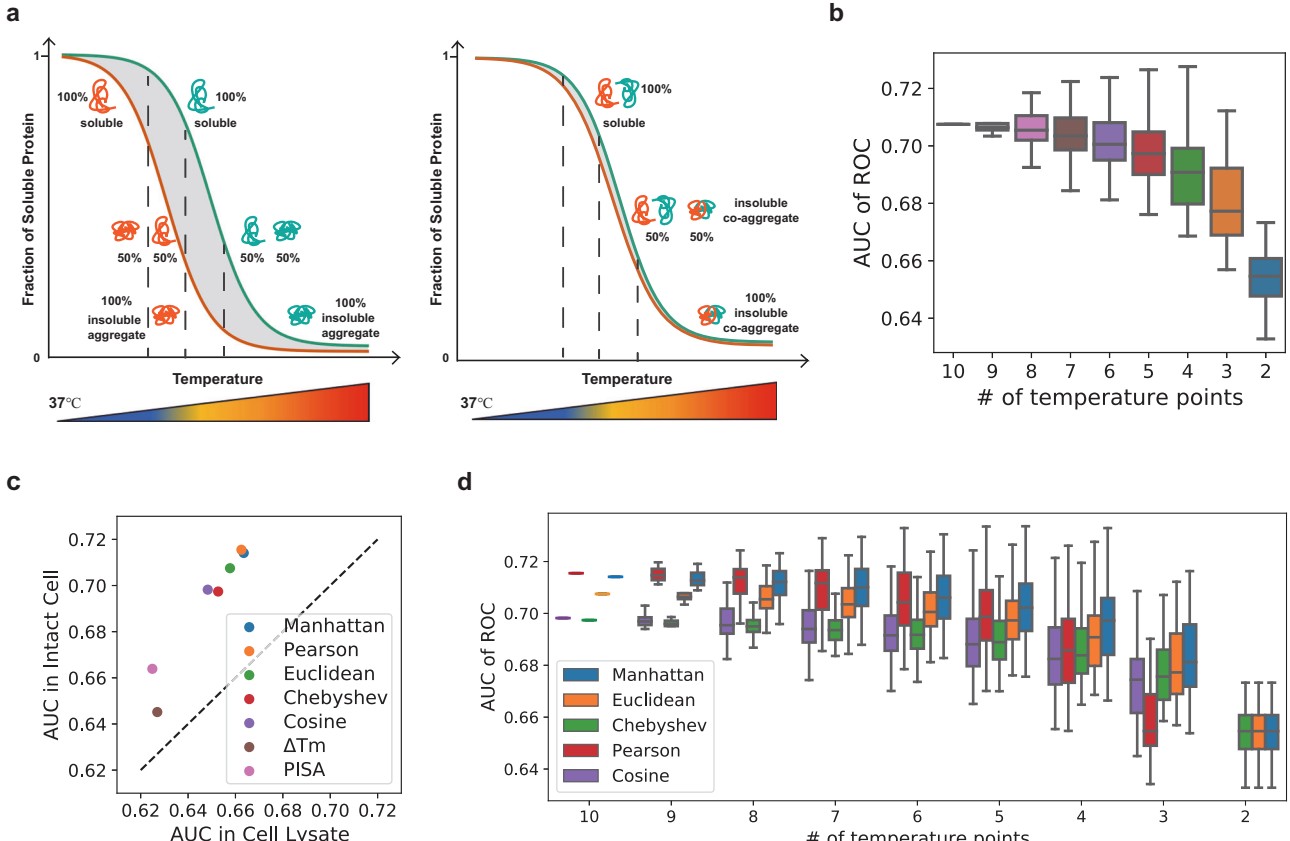

**Fig. 1 | Distance measurements evaluated in Slim-TPCA. a** Principle of Slim-TPCA for monitoring protein–protein interactions based on similarity between protein melting curves. In Slim-TPCA, lesser evenly spaced temperature points are used instead of the 10 gradient temperature points in the conventional method. **b** Box plot of predictive power of TPCA signature quantified by Euclidean distance with different number of temperature points for differentiating between interacting and non-interacting protein pairs. AUC: Area Under Curve; ROC: Receiver Operating Characteristic curve. The box extends from the lower quartile to the upper quartile values of the data, with a line at the median. When *n* out of 10 temperature points are selected, there are $10!/(10 - n)!n!$ unique temperature point combinations. All

combinations of temperature points are tested and predictive power generally decreases with less temperature points. **c** Predictive power of TPCA signature quantified by different measures across 10 temperature points for differentiating between interacting and non-interacting protein pairs. Tm Melting Temperature, PISA Proteome Integral Solubility Alteration. **d** Box plot of the ability of different measures in predicting PPI when used with fewer temperature points in the intact cell data. Each box plot represents the distribution of AUC obtained for all unique combi*n*ation sets for *n* out of 10 temperature points using different distance measurements. Pearson distances and Cosine distances cannot be computed with 2 temperature points. Source data are provided as a Source Data file.

fewer temperature points with slight loss in predictive power. Nevertheless, we note that while the decrease in AUC is marginal with fewer temperature points, the difference can still be substantial in the prediction of protein–protein interactions when considering large number of proteins and possible protein pairs. Here, we had used PPI prediction to evaluate information content embedded in the full temperature set and its subsets. For the identification of modulated protein complexes, which we advocate the use of TPCA for, using less temperatures permit samples from multiple experimental conditions and replicates to be analyzed concurrently on MS instrument using isobaric labeling reagents e.g., TMT. This is important to eliminate batch variation in machine measurement and ensure the same peptides are quantified across samples to reduce false positives and facilitate downstream analysis.

Next, we analyzed which distance/similarity measures are most suitable for use with fewer temperature points. We analyzed 6 additional distance/similarity measures namely: Manhattan distance, Chebyshev distance, Cosine distance, Pearson's correlation coefficient, PISA (area under melting curves), and $\Delta T_m$ (melting temperatures). We observed the AUC values using 10 temperature point data with Manhattan distance and Pearson's correlation coefficient are higher than that using Euclidean distance for both cell lysate and intact cell data (Fig. 1c and Supplementary Fig. 1). However, the performance

of Pearson's correlation coefficient deteriorates rapidly with fewer temperature points (Fig. 1d and Supplementary Fig. 2) compared to using Manhattan distance and Euclidean distance. Overall, Manhattan distance consistently gives best performance with few temperature points (Fig. 1c, d) and is used in our subsequent analysis.

## Temperature points selection in Slim-TPCA

Next, we identified the combination of 2 and 3 temperature points (excluding 37 °C) that produce the highest correlation with 10 temperature points for random protein pairs using Manhattan distance. Here, we assume that better correlation indicates higher coherency in the TPCA signature embedded in the data. Our analysis revealed that 37 °C, 49 °C and 58 °C are the most suitable for 3 temperature points combination (Fig. 2a and Supplementary Fig. 3), while 37 °C, 46 °C, 55 °C and 61 °C are most suitable for 4 temperature points combination (Fig. 2b). We then used the identified combination of temperature points to analyze another set of TPCA data obtained from K562 cells which were treated with methotrexate (MTX). Previously, we performed bootstrapping to estimate the statistical significance of TPCA signature (non-random similarity among melting curves of proteins) observed in CORUM complexes. Here, we performed similar analysis using Manhattan distance comparing the TPCA signatures derived with 3 and 4 temperature points versus 10 temperature points.

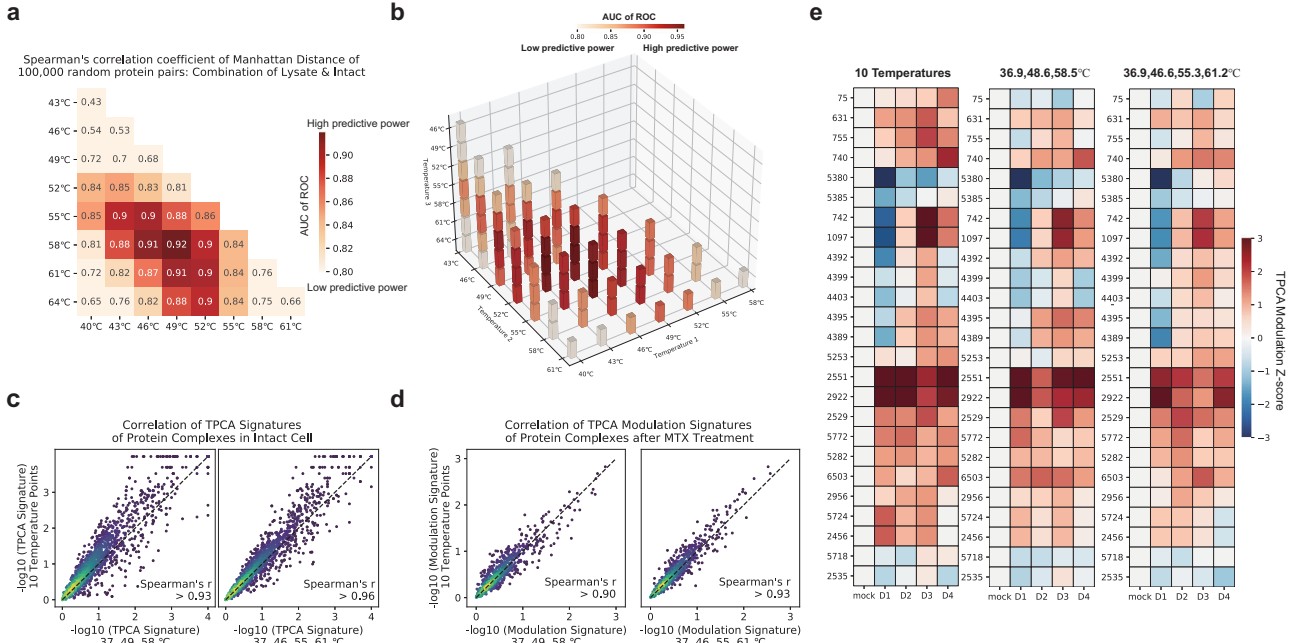

**Fig. 2 | Selection of optimal combinations of temperature points for TPCA profiling. a** Correlation of Manhattan distances for 100,000 random protein pairs calculated using 3 temperature points and 10 temperature points. The combinations of 3 temperature points $\in$ {37 °C, $T_a$ °C, $T_b$ °C}, where $T_a$, $T_b$ $\in$ {40 °C, 43 °C, 46 °C, 49 °C, 52 °C, 55 °C, 58 °C, 61 °C, 64 °C}. **b** Correlation of Manhattan distances for 100,000 random protein pairs calculated using 4 temperature points with those calculated using 10 temperature points. The combinations of 4 temperature points $\in$ {37 °C, $T_a$ °C, $T_b$ °C, $T_c$ °C}, where $T_a$, $T_b$, $T_c$ $\in$ {40 °C, 43 °C, 46 °C, 49 °C, 52 °C, 55 °C, 58 °C, 61 °C, 64 °C}. **c** Statistical significance of TPCA signature for CORUM complexes ($p$-value) quantified with 3 and 4 temperature points as compared to 10 temperature points ($n$ = 960 complexes). **d** Statistical significance of dynamic (modulated) TPCA signatures for CORUM complexes (TPCA Modulation Signature) quantified with 3 and 4 temperature points as compared to 10 temperature points ($n$ = 821 complexes). **e** Variation in TPCA signatures for identified modulated protein complexes during virus infection. The statistical significance obtained with 10 temperature points (left[23]) can basically be recapitulated using 3 temperature points (middle, 36.9 °C, 48.6 °C, and 58.5 °C) and 4 temperature points (right, 36.9 °C, 46.6 °C, 55.3 °C, and 61.2 °C). The figure for 10 temperature points is as obtained from the original publication[23] where the color scheme for the 3 and 4 temperature points are matched as close as possible. Source data are provided as a Source Data file.

Encouragingly, the statistical significance of TPCA signatures of the complexes in the CORUM database calculated using our selected 3 and 4 temperature points are highly similar to those calculated using 10 temperature points with Spearman's $r$ greater than 0.93 and 0.96 for 3 and 4 temperature points respectively (Fig. 2c). Next, we assessed the ability of the identified combination of temperature points to identify protein complexes dynamically modulated by MTX treatment. Similar to previous work, the statistical significance of the difference in TPCA signature (termed TPCA Modulation Signature) across two conditions was estimated with bootstrapping analysis. We also observed high correlation in statistical significance estimated with 10 temperature points to that estimated with 3 and 4 temperature points with Spearman's $r$ greater than 0.90 and 0.93 respectively (Fig. 2d and Supplementary Fig. 4). Thus, statistical significance estimated using our identified combinations of 3 or 4 temperature points are in good agreement with those calculated using 10 temperature points.

To further investigate the coherence of the identified combinations of temperature points across different experimental data, we used the data published by ref. 23 to validate the selection of temperature points (Supplementary Data 3). In the work, the authors applied TPCA profiling to study the dynamics of protein complexes and identified a subset that is modulated during human cytomegalovirus infection. Using Manhattan distance and similar sets of temperature points, the dynamics of identified modulated complexes across 4 days of infection obtained by 10 temperature points (Fig. 2e, left panel) are generally consistent with the results calculated using 3 temperature points of 36.9 °C, 48.6 °C, 58.5 °C (Fig. 2e, middle panel) and 4 temperature points of 36.9 °C, 46.6 °C, 55.3 °C, and 61.2 °C (Fig. 2e, right panel and scatter plot in Supplementary Fig. 5). This

demonstrates that the identified combinations of temperature points are generally applicable across samples from different experiments.

Importantly, the identified optimal combination of temperature points is in line with our expectation that picking the intermediate, widely separated temperature points that are evenly distributed around average melting temperature ($T_m$) will maximally encapsulate TPCA signal embedded across the temperature range tested. To test this theory, we chose TPP/MS-CETSA data from Arabidopsis thaliana[39], whose plant proteome has a lower melting temperature than the mammalian proteome. Similarly, the combination of temperatures producing data that correlate most with that from 10 temperature data (Supplementary Fig. 6) is also intermediate, widely, and equally distributed around the average melting temperature of the proteome of Arabidopsis thaliana (~46.6 °C). Furthermore, we also analyzed datasets from 15 species housed in the Meltome database[40] that include B. subtilis and C. elegans etc. The data for these species similarly demonstrate that the right combination of temperature points in the Slim-TPCA method correlates well with data derived from full temperature set (Supplementary Fig. 6). These results suggest the framework proposed in this work could also be adopted for other species. For other species, we recommend to first perform conventional TPCA experiments to determine the melting temperature and to verify the feasibility of the Slim-TPCA method in that species.

Using Slim-TPCA with 3 or 4 temperature points, we can now analyze samples under different conditions or multiple sets of experimental conditions, such as different treatment duration using a set of TMT reagents. As such, Slim-TPCA has the advantage of being able to reduce sample consumption, reduce batch effects, and provide more flexibility in experimental design.

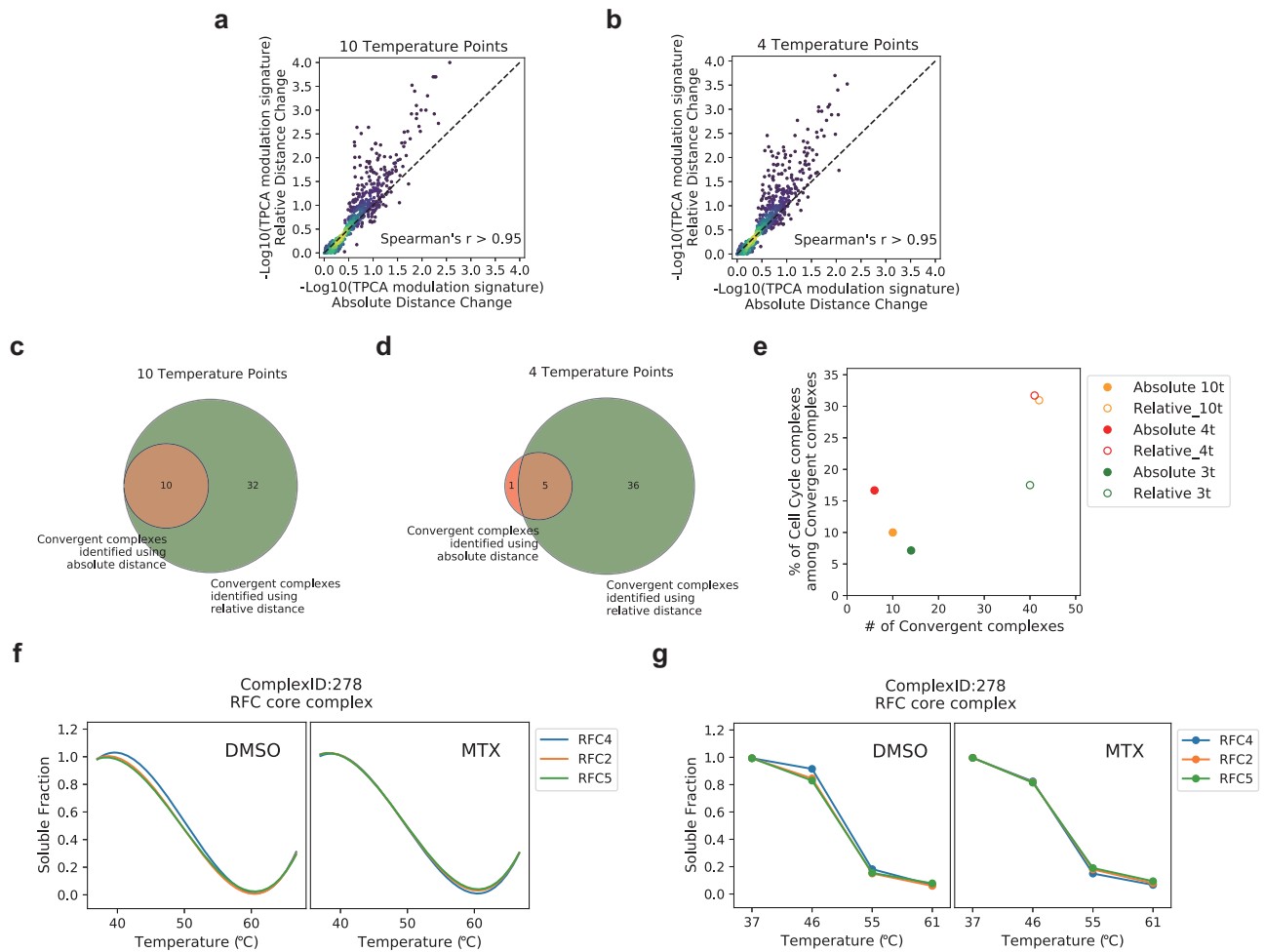

**Fig. 3 | Relative distance algorithm identifies more dynamic complexes in TPCA and Slim-TPCA. a**, **b** Correlation of the TPCA Modulation Signature of CORUM complexes obtained using the relative distance algorithm and the absolute distance algorithm for 10 and 4 temperature points, respectively (*n* = 821 complexes). **c**, **d** Comparison of the dynamic complexes identified by the relative distance algorithm with the absolute distance algorithm at 10 and 4 temperatures,

respectively. **e** Number and functional class of convergent protein complexes identified with different algorithms. Compared to the absolute distance algorithm, the relative distance algorithm identifies more dynamic complexes associated with cell cycle, according to the annotation in the CORUM database. **f**, **g** Curves of subunits in Complex 2792 under DMSO and MTX, with 10 temperature points and 4 temperature points, respectively. Source data are provided as a Source Data file.

## Relative distance algorithm optimization to detect dynamic complexes

The TPCA method has the capacity to identify protein complexes modulated across conditions based on changes in TPCA signature. Specifically, to detect dynamically modulated protein complexes, the average difference in solubilities between subunits in a protein complex is compared across conditions and benchmarked against 10,000 randomly generated protein sets of the same size. Protein complexes that experience more extreme variations than artificially generated protein sets are considered as dynamically modulated protein complexes, based on computed TPCA Modulation Signature which essentially is estimated p-value of observed difference. In the original TPCA method, the variation is obtained by direct subtraction of their average distances under different conditions, which we termed as the absolute distance algorithm. This absolute distance algorithm has been demonstrated to detect known activated complexes, such as the CAF-1 complex formed at the replication fork under MTX treatment which trapped K562 cells in S-phase of cell cycle. Nevertheless, the absolute distance algorithm can miss modulated complexes that already has strong TPCA signature in a reference condition but are further strengthened in another condition. For example, we will expect a population of cells to be in different phases of cell cycle with associated protein complexes exhibiting certain degree of TPCA signature.

While the stoichiometry of such complexes could further peak, changes in the absolute distance are limited given the relatively strong TPCA signature of these complexes at basal conditions. In addition, proteins could also have similar thermal solubility independent of each other that are enhanced by interaction.

To detect protein complexes in above-mentioned situations, we introduce a new scoring algorithm based on relative distance to identify such complexes (see Methods). Specifically, changes in TPCA signature are quantified as a ratio of basal TPCA signature. We evaluate this relative distance algorithm using the sets of data from previous work where K562 cells enriched in S-phase of cell cycle using MTX are compared to non-synchronized cells. First, using the same method of sampling to generate random protein sets, we analyzed the correlation between the TPCA Modulation Signature obtained by the relative distance algorithm and the absolute distance algorithm. The TPCA Modulation Signature of CORUM complexes calculated using the relative distance algorithm maintains good consistency with those calculated using the absolute distance algorithm for both 10-temperature TPCA and 4-temperature Slim-TPCA method (Fig. 3a, b). Importantly, insignificant (high) p-values are highly consistency across absolute and relative distance algorithms but the latter results in enhanced p-value for a subset of complexes that already have low p-value from absolute distance scoring. In other words, the relative

distance algorithm neither lead to depreciation of *p*-value estimated nor enhancement for those with very insignificant *p*-value but identified a subset of complexes with enhanced *p*-value indicating increased sensitivity.

As expected, the relative distance algorithm identified more modulated protein complexes than absolute distance algorithm across 10, 4 and 3 temperatures tested. In particular, the modulated protein complexes identified by relative distance scoring algorithm encompassed all complexes identified by absolute distance scoring algorithm with 10 temperature points (Fig. 3c), and all but one complexes when using 4 temperature points (Fig. 3d). Importantly, the relative distance algorithm also identified a higher percentage of protein complexes associated with cell cycle and DNA processing (according to the annotation provided by the CORUM database) which is more in accordance with the experimental condition in which samples are collected where cells are trapped in the S-phase of cell cycle (Fig. 3e). For example, the RFC complex, which is involved in DNA replication, is identified to be dynamically modulated with the relative distance algorithm but is missed by absolute distance algorithm (Fig. 3f, g). This suggests that the relative distance algorithm has improved sensitivity in identifying modulated protein complexes.

### Optimization of *P*-value algorithm using fitted Beta distribution

To quantify the significance of TPCA signature observed as well as its changes across conditions, we had adopted a bootstrapping approach to estimate *p*-value which is the probability of observing random complexes with similar or more extreme values (either absolute TPCA signature or changes). Specifically, the bootstrapping (sampling) algorithm compares the average distance or change in average distance of the complex with 10,000 randomly sampled protein sets of the same size. This method obtains discrete *p*-values and takes up considerable computing time. Here, we optimized this part of TPCA workflow fitting a specific data distribution with smaller sample size for estimating *p*-value.

We validated this approach with the intact cell data of K562 published in the proof-of-concept paper with 4 temperature points of 37 °C, 46 °C, 55 °C, and 61 °C recommended earlier. First, using the Chi-square statistics, we observed that Beta distribution generally fitted well to the distributions of Manhattan distance derived from bootstrapping algorithm across different number of temperatures for both absolute and relative distance scoring (Supplementary Fig. 7). Next, we verified that Beta distribution fitted with 100, 200, 500, and 1000 random complexes are similar to actual distribution of Manhattan distance, Δ absolute distance, and Δ relative distance computed from 10,000 random complexes, indicating that a small number of samples fitted to Beta distribution can be used in place of excessive sampling (Fig. 4a–c). Thus, we tested the Beta distribution fitting algorithm with the dataset of K562 cells trapped in S-phase of cell cycle using MTX. We observed more dynamically modulated protein complexes being identified with this approach, which is generally accompanied with a slight increase in the proportion of protein complexes related to cell cycle and DNA processing across 10, 4 and 3 temperature points (Fig. 4d, e).

Subsequently, we cross-validated the new algorithm on the viral infection dataset with 4 temperature points of 36.9 °C, 46.6 °C, 55.3 °C, and 61.2 °C. We identified the complexes with significant TPCA signatures or differentially modulated during viral infection and compared them with the complexes identified with the original sampling method. The new algorithm of fitting Beta distributions essentially recovered all the protein complexes identified by the traditional sampling algorithm (Fig. 4f–h) regardless of whether Manhattan distance, difference in absolute Manhattan distance or difference in relative Manhattan distance is used. In addition, we also identified new modulated protein complexes reportedly involved in viral infections such the exocyst complex regulating vesicular trafficking[41], the RNase

MRP complex and mRNA decay complex involved in viral RNA degradation[42,43]. Importantly, this algorithm requires less than 1% of the time needed in the original sampling processing with 10,000 random complexes.

### Slim-TPCA identifies dynamically modulated complexes in response to glucose deprivation

Cancerous cells are often found to have altered energy metabolism where ATP is preferably generated from less efficient glycolysis in cytoplasm over oxidative phosphorylation pathway in mitochondria. Glucose, a key energy source that is supplied abundantly in most culture media, is consumed excessively, and found to induce cell death in many cancerous cells when withdrawn (glucose addiction)[44]. However, not all cancerous cells exhibit dependency on glucose for survival and proliferation. Glutamine, another key nutrient consumed excessively by cancerous cells, could be a source for ATP production through its α-ketoglutarate intermediate that feeds into the tricarboxylic acid (TCA) cycle[45]. Here, we incorporate all the modifications and enhancements made in Slim-TPCA to investigate the dynamics of CORUM-annotated protein complexes under glucose deprivation which we applied on K562 suspension cells to minimize perturbation to cell morphology and cellular physiology during cell harvesting (Fig. 5a Supplementary Figs. 8–15).

First, we observed minimal apoptosis in K562 cells at different duration of glucose deprivation tested based on FACS analysis (See Supplementary Methods & Supplementary Figs. 16–17). Nevertheless, replication of cells was stalled in the first 24 h with resumption observed after 48 h of glucose deprivation, indicating successful adaptation of K562 cells to glucose-deficient condition. Accordingly, K562 cells were harvested at 0th, 4th, 8th, 24th and 48th h of glucose deprivation. At each time point, vials of cells were subjected to the optimal set of 37 °C, 46 °C and 55 °C identified previously, and processed with TMT16 reagents for concurrent analysis on MS. Three sets of biological replicates are performed where 6476, 6627 and 6411 proteins were identified respectively with 8311 unique proteins identified altogether (Supplementary Data 4). We observed high data reproducibility with average Pearson's correlation of 0.76 and 0.82 in protein solubility at each time point for 46 °C and 55 °C respectively and low CV value across the three biological replicates (Supplementary Figs. 18–20). Average values across replicates are computed for a total of 5813 proteins that appear in at least 2 replicates and survived filtering criteria (see Materials & Method) which are used for downstream analysis.

TPCA signature permits the identification of both increasingly associated (convergent) and increasingly dissociated (divergent) protein complexes in situ compared to a baseline. Here, glucose-deprived K562 cells are compared to cells at on start of glucose deprivation (0th h). A total of 783 complexes from the CORUM database were investigated. On average, we observed about twice more divergent than convergent protein complexes across all the time points profiled (Fig. 5b), arguably in accordant with the downregulation expected for most basal cellular processes. Indeed, we observed proteins in divergent protein complexes over-represented in translation and ribosome biogenesis from samples at 4th, 8th, 24th but not at 48th h of glucose deprivation (Fig. 5c and Supplementary Fig. 21).

Thus, we inspected the previous observation more closely with the TPCA profile of 40S and 60S ribosomes at different time points of glucose deprivation. The thermal solubilities of both 40S and 60S ribosomal proteins are most dissimilar to each other (indicating dissociation of complexes based on TPCA theory) at 4th, 8th, 24th h of glucose deprivation. However, the TPCA profile of ribosomal proteins at 48th h of glucose deprivation is highly similar to data collected from samples without glucose deprivation suggesting resumption of translational activities. Previous studies had revealed system-wide attenuation in protein translation under glucose deprivation[46,47] that

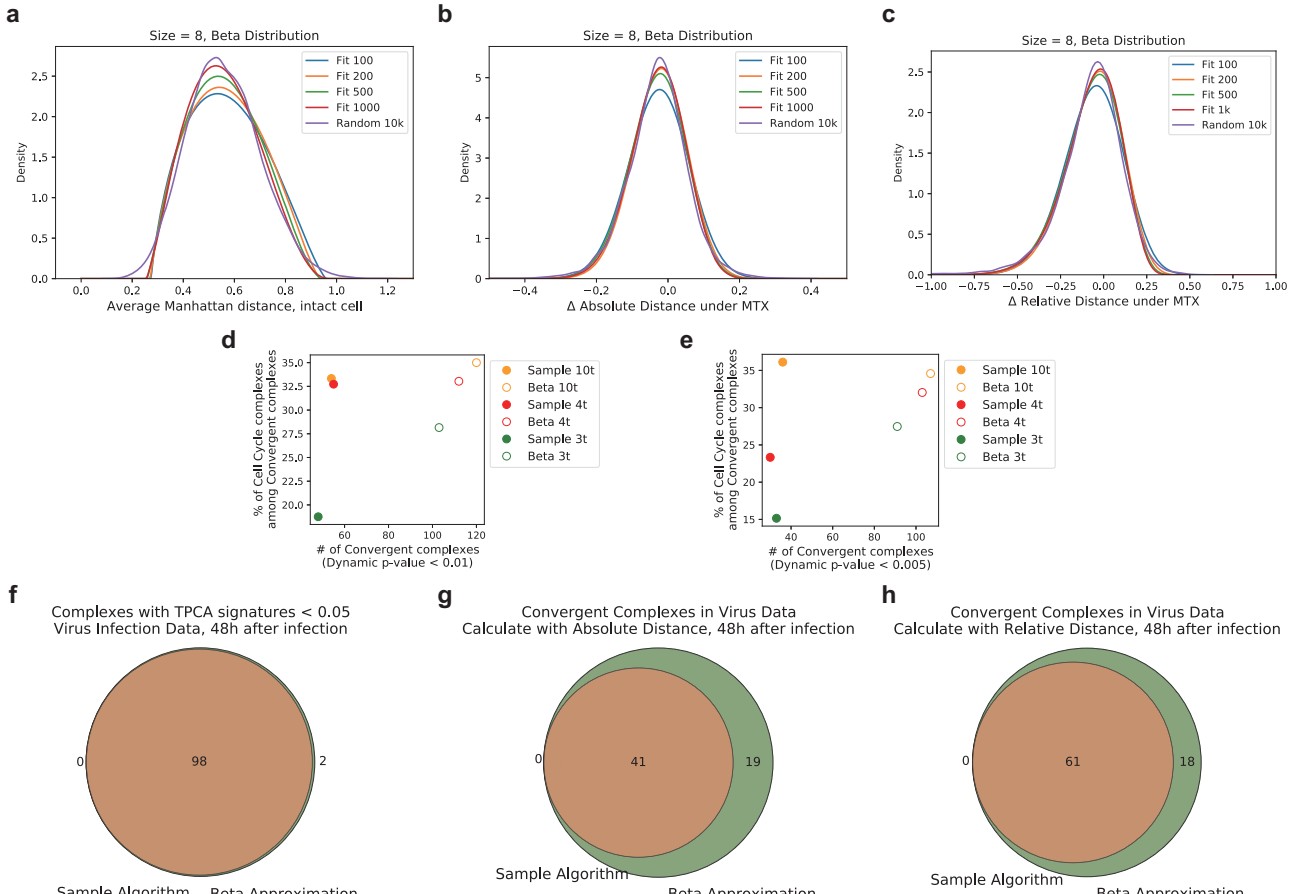

**Fig. 4 | Optimization of P-value estimation algorithm using fitted Beta distribution.** The algorithm with small number of samples combined with the fitted Beta distribution can simulate the distribution of large number of samples very well as observed for Manhattan distance (**a**), Δ absolute distance (**b**) and Δ relative distance (**c**). **d**, **e** Number and functional class of convergent protein complexes identified based on new *P*-value estimation algorithm. Beta distribution fitting algorithm identifies more dynamically modulated complexes with higher percentage associated with cell cycle and DNA processing generally. The result of virus replication-related complexes with TPCA signatures and TPCA Modulation Signature obtained by the new algorithm maintain a good agreement with those obtained by the traditional algorithm with massive sampling based on Manhattan distance (**f**), Δ absolute distance (**g**) and Δ relative distance (**h**). Source data are provided as a Source Data file.

occurred independent of mRNA abundance[48]. Thus, the disassembly of ribosomes could possibly contribute or arise from decreased protein translation, an energy-intensive process[49,50] which prerequisite in many cells is the availability of glucose[47]. Interestingly, we also observed similar pattern of divergent and convergent TPCA signatures for the 28S and 39S ribosomal subcomplexes localized in mitochondria (Fig. 6a) suggesting cell-wide attenuation of protein translation. This demonstrated the utility of the streamlined TPCA protocol for identifying modulated protein complexes including those localized in membrane-bound organelles.

Intuitively, the deprivation of glucose could potentially alter mitochondrial protein complexes involved in cellular respiration. Thus, we focus on validating our streamlined TPCA protocol on such protein complexes annotated in the CORUM database. On average, less than 20% of a core set of CORUM-curated protein complexes are found modulated at any time point profiled. Encouragingly, most mitochondrial protein complexes involved in respiration (consisting of respiratory chain complex V, the various subcomplexes and intermediates of respiratory chain complex I) exhibit differential TPCA signature under glucose deprivation.

In particular, the respiratory chain complex V (annotated as F1F0 ATPase complex in CORUM database) exhibits very consistent divergent TPCA signature that is strongest in the first 8 h but weaken thereafter. Based on TPCA profile, there seem to be a recovery of the complex sometime after 24 h of glucose deprivation (Fig. 5d). On the other hand, the various intermediates and subcomplexes of respiratory chain complex I exhibit more diverse and muted but nevertheless statistically significant changes in TPCA signatures. Interestingly, while most of these intermediates and subcomplexes manifest divergent TPCA signatures that peaked at 24th h, the TPCA signature was reversed (convergent) at 48th h. Similarly, divergent TPCA signature was observed for the MIB (mitochondrial intermembrane space bridging) complex (Fig. 6a) formed between the MICOS (mitochondrial contact site) complex and the SAM (sorting and assembly machinery) complex located in the inner and outer membrane respectively[51,52]. These complexes are involved in the formation of cristae and junctions that optimize the microenvironment for assembly of respiratory chain complexes and ATP production. Like the various intermediates and subcomplexes of respiratory chain complex I, the TPCA signature of the MIB complex also diverged in the first 24 h of glucose deprivation but subsequently converged based on samples collected at 48th hour of glucose deprivation.

Overall, TPCA profiling suggests that cellular respiration is muted in the first 24 h with sign of recovery thereafter based on samples collected at 48th h of glucose deprivation. To investigate this further, we quantified the amount of ATP in K562 cells at different time point of glucose deprivation. Somewhat unexpected initially, we observe small but gradual increase in total ATP and average ATP abundance (per cell)

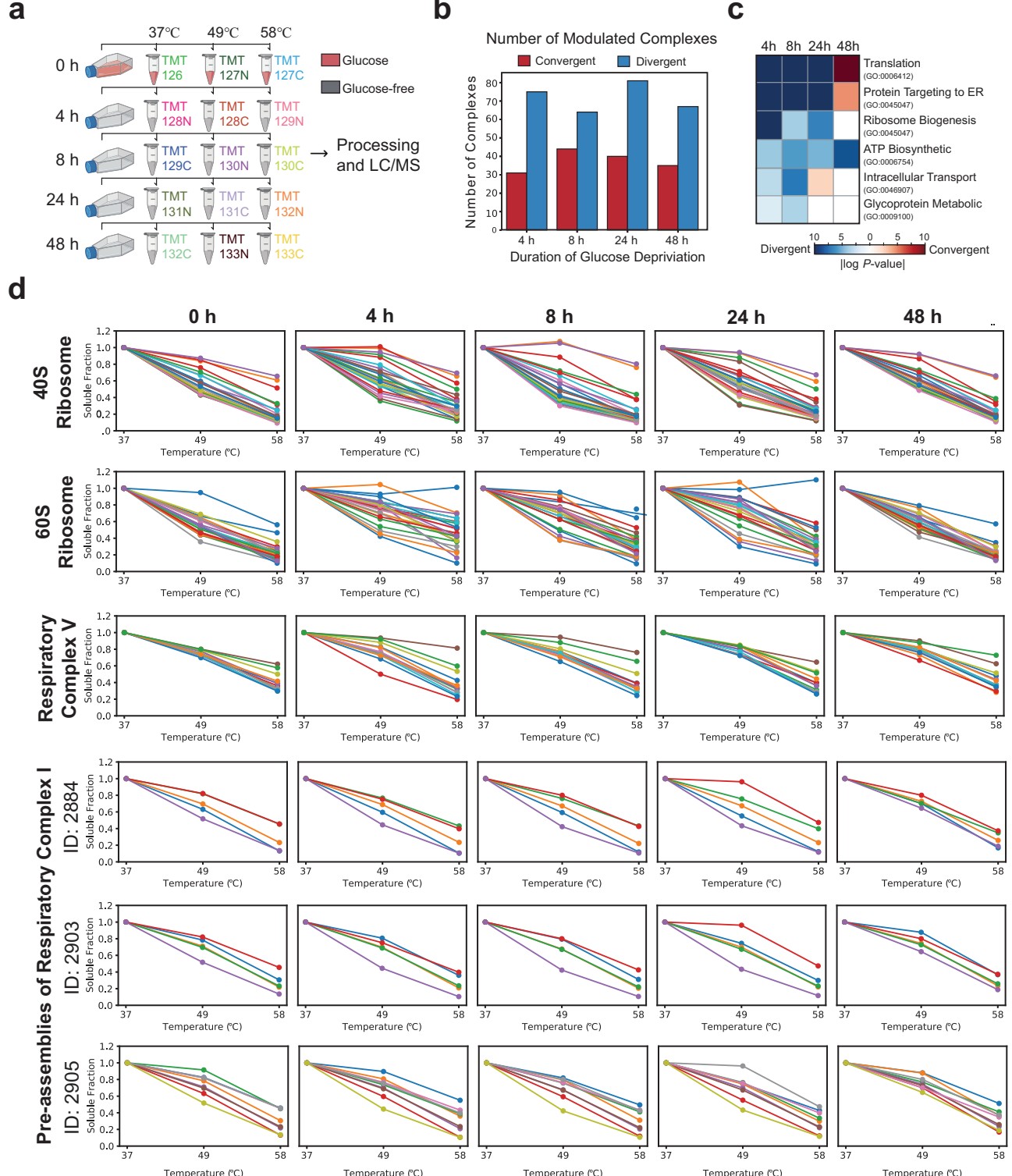

**Fig. 5 | Application of optimized TPCA profiling to K562 cells under glucose deprivation. a** Schema of samples collected. K562 are deprived of glucose in culture medium and harvested at different time point for heating at 3 temperatures for TPCA profiling. **b** Numbers of CORUM-annotated protein complexes detected at different time point with convergent and divergent TPCA signature. **c** Major functional classes of proteins in divergent protein complexes. **d** TPCA profile of selected convergent and divergent protein complexes at different time point of glucose deprivation. Source data are provided as a Source Data file.

in the first 24 h of glucose deprivation with a sharp decrease in ATP in cells collected at 48th h of glucose deprivation (Fig. 6b). Thus, ATP abundance seems to correlate inversely with TPCA signatures of the various respiratory complexes. We reasoned this could arise from the huge ATP demand of cells in many basal cellular activities after

successful adaptation to glucose-deficient condition. Indeed, this seems to be supported by the observed convergent of 40S and 60S ribosomal protein complexes at 48th h of glucose deprivation suggesting increased protein translation which arguably consumes a large majority, if not most, of cellular ATP[49,53]. Overall, we observed that

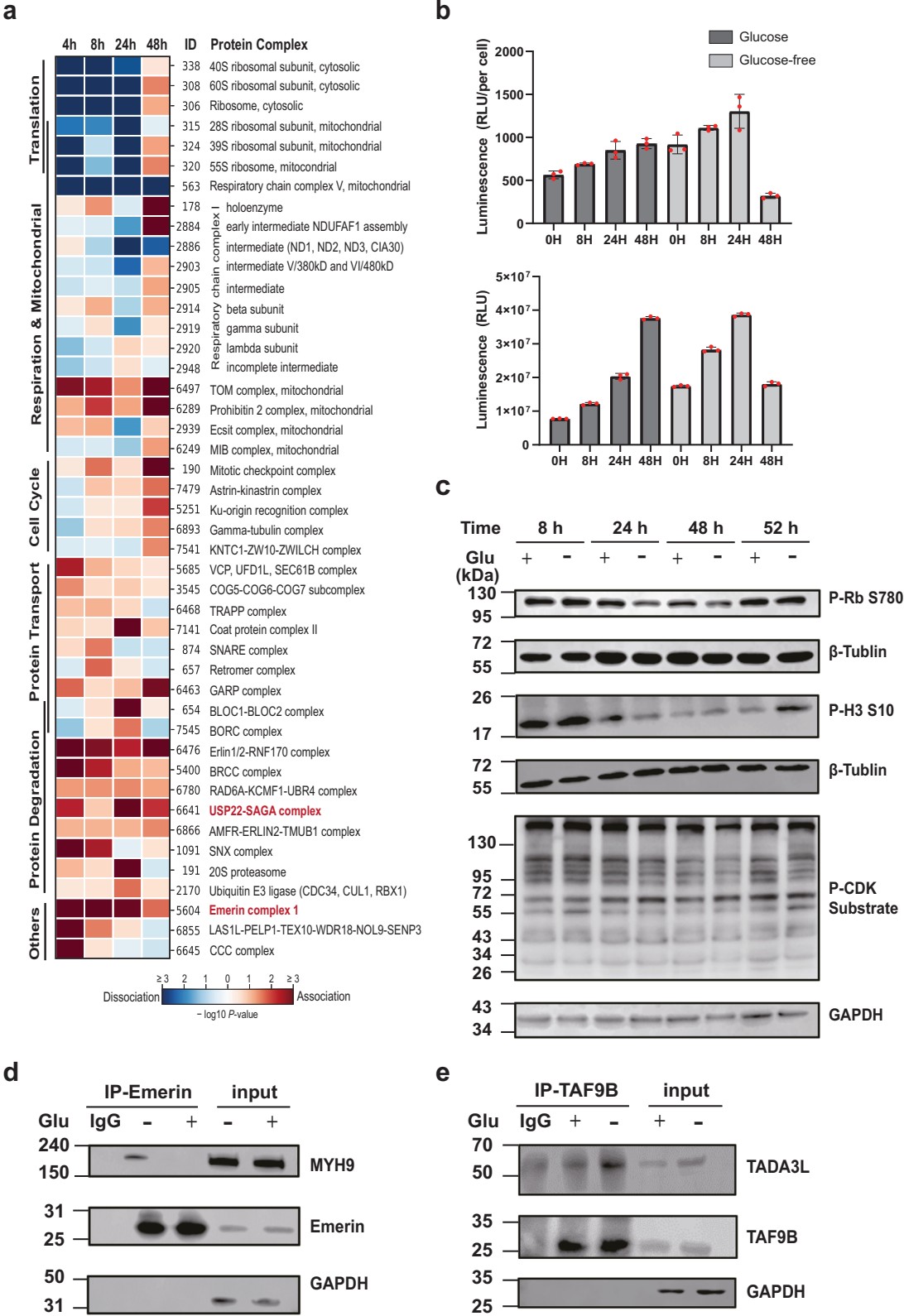

when ribosomal assembly is presumably down-regulated (based on TPCA signature), higher cellular ATP abundance was observed.

As most of the identified modulated protein complexes dissociated and are involved in basal cellular processes that we will reasonably expect in cells deprived of a source of energy, we focus on analyzing convergent protein complexes to identify those potentially implicated in metabolic reprogramming of cells in the absence of

glucose. While the various mitochondrial respiratory complexes mostly exhibit divergent TPCA signatures, other differentially modulated mitochondrial protein complexes manifest a strong convergent TPCA signature instead. They are the TOM complex involved in mitochondrial protein import, the Prohibitin 2 complex that inhibits caspase-dependent apoptosis[54], and the ECSIT (evolutionarily conserved signaling intermediate in Toll pathways) complex

**Fig. 6 | Glucose deprivation induces protein complexes changes in K562 cells.**
**a** Dynamics of CORUM-annotated protein complexes under glucose deprivation as
profiled by Slim-TPCA. Complexes are grouped based on their functions. The
complexes marked in red were selected to do Co-IP experiment. **b** K562 cells were
incubated in presence and absence of glucose followed by quantification of ATP
based on luminescence intensity ($n = 3$ independent biological experiments). Data
are presented as mean values ± s.d. (error bars). **c** K562 cells were incubated in
presence and absence of glucose, harvested at different time points and analyzed

by immunoblotting using Phospho-Rb (Ser780) antibody, Phospho-Histone
H3(Ser10) antibody, and Phospho-CDK Substrate antibody. GAPDH antibody and β-
Tubulin antibody were used as loading control ($n = 3$ independent biological
experiments). **d, e** K562 cells were incubated in presence and absence of glucose
for 48 h, and analyzed by Immunoprecipitation and immunoblotting using MYH9
antibody, Emerin antibody, TAF9B antibody and TADA3L antibody. GAPDH anti-
body was used as loading control ($n = 3$ independent biological experiments).
Source data are provided as a Source Data file.

implicated in the regulation of mitochondrial respiration and
mitophagy[55] (Fig. 6a).

In addition, we observed increased association of many protein
complexes involved in protein transport and vesicle formation with
peaked convergent TPCA signatures observed at 8th and 24th h time
points sampled that included BLOC and BORC protein complexes
associated with lysosome. At the same time, we observed convergent
protein complexes implicated in degradation of various proteins
throughout the time points profiled. These complexes include the
Erlin1/2-RNF170 complex and the SNX complex implicated in degra-
dation of IP3 (Inositol 1, 4, 5-trisphosphate) receptors and EGFR (epi-
dermal growth factor receptor) respectively, as well as various protein
complexes that possess E3 ligases (e.g., the BRCC complex and the
AMFR-ERLIN2-TMUB1 complex). Correspondingly, we observed con-
vergence of 20S proteasome and an E3 ligase complex (consisting of
CDC34, CUL1 and RBX1) that peaked at 24th h of glucose deprivation.
Hence, there seems to be an increase in protein degradation activities
perhaps to free amino acids for energy metabolism.

We also observed a few protein complexes that converge only at
48th h and are implicated in chromosome segregation and mitosis.
These complexes include both the mitotic checkpoint complex, and
the astrin-kinastrin complex that is involved in correct alignment of
chromosome during mitosis, the gamma-tubulin complex (CORUM ID:
6893) that associates with centrosome involved chromosome segre-
gation, and the KNTC1-ZW10-ZWILCH complex that is associated with
kinetochores on chromatids. Collectively, the convergent of these
complexes suggests possible onset of mitosis around 48th h of glucose
deprivation. As such, we performed immunoblotting for specific
mitotic protein markers namely phosphorylation of histone H3 and
retinoblastoma protein at S10 and S780 respectively. We observed
decrease of these markers at 24th and 48th h but an increase at 52nd h
of glucose deprivation, suggesting the protein complexes implicated
in chromosome alignment and segregation converge before activation
of mitosis-associated signaling. We also compared the phosphoryla-
tion levels of CDK substrates, another mitotic protein marker, by
immunoblotting. Similar to what was observed for phosphorylation of
histone H3 and retinoblastoma protein, we observed reduced CDK
substrate phosphorylation before 48th h of glucose deprivation that
increased at the 52nd h indicating possible onset of mitosis (Fig. 6c).

Finally, we performed Co-IP to experimentally validate two con-
vergently modulated protein complexes identified (marked red in
Fig. 6a). One is Emerin complex 1 which function to organize the
nuclear membrane during cytokinesis[56], and the other is USP22-SAGA
complex which function is a regulatory center for signaling, chromatin
modification, DNA damage repair, and gene control[57]. The extent of
TPCA signals of these two complexes at 24 h and 48 h of glucose
deprivation was relatively consistent (Supplementary Fig. 22). Slim-
TPCA analysis revealed increased association between subunits for
both Emerin complex 1 and USP22-SAGA complex during glucose
deprivation which was recapitulated in our Co-IP experiments
(Fig. 6d). After 48 h of glucose deprivation, the TPCA signals of the
proteins used for experimental validation converged gradually.

## Discussion

Protein complexes are the key effectors in most cellular activity that
need to be dynamically assembled and dissolved according to cellular

needs. Unlike most proteome-wide experimental methods, the intra-
cellular assembly state of protein complexes can be profiled in situ
with TPCA rapidly. The method had been verified in primary cell and
tissues and been deployed to identify protein complexes modu-
lated during cell cycle, viral infection and T-cell infection. Con-
ceptually, the method could be readily deployed across various
cellular conditions to profile the assembly state of uncharacterized
protein complexes identified from large scale projects to aid in their
functional annotation.

Here, we introduce Slim-TPCA, an improved TPCA method using
fewer temperature points and improved data deconvolution algo-
rithms offering a series of improvements in terms of experimental
design and data analysis over previous method. We showed that less
temperature points can be used with minimal or negligible lost in TPCA
signal. This permits more flexibility in the experimental design,
allowing more time points or experimental conditions per analysis
while reducing sample needed, experimental cost and batch effect in
MS analysis. At the same time, the new data processing algorithms
offer over 100X reduction in time required for statistical calculations
while identifying modulated protein complexes with improved
sensitivity.

In this work, we had identified the optimal smaller sets of tem-
peratures for TPCA profiling of mammalian cells. The temperature
points identified fitted well with the expectation that reasonably
spaced temperatures maximally captured TPCA signature. For the
application of TPCA profiling in other species, similar principle could
be applied to the selection of optimal set of temperatures. We had also
adopted Manhattan distance for encapsulating TPCA signal instead of
Euclidean distance used in the original TPCA work as Manhattan dis-
tance gave the best overall performance across all the number of
temperatures tested. Although Pearson's correlation performs as well
as Manhattan distance, its ability to encapsulate TPCA signature
dimmish more drastically with less data points compared to Man-
hattan distance and other metrics. We speculate frequency or occur-
rence rate of protein–protein interactions likely scale linearly with
TPCA signal, hence is modeled better with Manhattan distance com-
pared with Euclidean distance measurements that amplify difference
non-linearly. Another major deviation from the previous TPCA format
is the use of relative change in TPCA signature over absolute difference
to better characterize the dynamics of protein complexes, especially
those with already strong inherent or basal TPCA signatures.

Conceptually, the use of relative distance to quantify changes in
TPCA signature across conditions can reduce false negative to identify
more modulated protein complexes but the approach can also lead to
a higher false positive rate. The latter is particularly more pronounced
if quality of the data is noisier, particularly when authentic biological
difference is smaller than intrinsic instrumental noise. In this case, the
use of error bars or standard deviation could guide interpreting
authenticity of difference observed. With Slim-TPCA, multiple repli-
cates can be analyzed simultaneously by MS instrument to at least
minimize variance in batch measurement. We previously reported that
precision or reproducibility in thermal solubility quantified depends in
part on the number of peptide-spectral match (PSM) and ion intensity
of proteins[13]. To reduce the false positive rate, proteins will less precise
values can be filtered based on these criteria. Alternatively, data from
multiple technical and biological replicates can be integrated to obtain

averages that are nearer to the true values. We prefer the latter approach as the former approach of filtering based on PSM and ion intensity can remove many proteins. Using data integrated from three biological replicates reported in previous work, we found the relative distance identified more modulated protein complexes that correlate with expected biological activities.

Another improvement made is over 100X reduction in time needed to compute the statistical significance of TPCA signature and its changes across conditions. Here, we adopted the strategy of using lesser sampling (500 rather than 10,000 sets of randomly selected protein) fitted to Beta distribution with computed p-value that correlate well with empirical p-value. This improvement is provided as an option in our provided Python package.

Finally, we combined all the improvements and changes made by applying Slim-TPCA to profile K562 cells at five time points of glucose deprivation using three temperature points which are performed in a set of TMT experiment for each replicate. Encouragingly, we identified modulated protein complexes involved in biological processes that are either reportedly or expectedly associated with glucose starvation. Our profiling identified increased dissociation of both cytoplasmic and mitochondrial ribosomes that is coherent with reported attenuation of protein translation during glucose starvation. Many mitochondrial protein complexes involved in or associated with respiration and complexes implicated in vesicle-based protein transport are found to be modulated revealing that TPCA is applicable to protein complexes localized within membrane-bound organelle and those associated with membrane.

In summary, we have incorporated multiple improvements for TPCA analysis of protein complex dynamics that include new statistical model and using a subset of temperatures. The latter permits multiple conditions and replicates to be analyzed concurrently on MS instrument using multiplexing reagents. This eliminates batch variation across multiple MS runs in the traditional TPCA analysis and reduces false positives. Nevertheless, the optimization using reduced number of temperatures could also involve trade-offs, for example, protein interactions that can be identified by traditional TPCA methods may be missed in Slim-TPCA, or vice versa, Slim-TPCA may identify false-positive protein interactions. Importantly, while we had quantitatively assessed the loss of information using subset of optimal temperatures to be marginal, the analysis is performed in the perspective of analyzing dynamics of protein complexes with TPCA. Thus, the findings and conclusions presented are not necessarily transferrable to other uses of protein thermal solubility data. In addition, it should be noted that detected changes in protein solubility, which presumably arise from changes in protein thermal stability, could result from other factors, including ligand/metabolite binding, post-translation modification and protein relocalization. TPCA specifically uses the hypothesized co-aggregation and co-precipitation of protein complexes to analyze only a subset of these changes. With users mindful of these, we envision Slim-TPCA and associated software package to expedite functional characterization of existing and newly identified protein complexes.

## Methods
### Cell culture
The K562 cell line was purchased from American Type Culture Collection (ATCC). K562 cells were cultured either in RPMI 1640 medium (Gibco) or glucose-free RPMI 1640 medium (Gibco) for 0 h, 4 h, 8 h, 24 h, and 48 h supplemented with 10% FBS (PAN) and 1% penicillin −streptomycin (Gibco) at 37 °C, and 5% $CO_2$ in a humidified environment. Cells were washed twice with ice-cold PBS prior resuspended at 150 μl PBS (Gibco). Each condition sample was distributed in parallel into three aliquots and subsequently heated in parallel in a PCR (VWR, Doppio Gradient) block for 3 min to the three temperatures (37 °C, 46 °C, 58 °C). Then, cell subjected to (2X) lysis buffer containing

concentration of 100 mM HEPES pH 7.5, 20 mM $MgCl_2$, 10 mM β-Glycerophosphate (sodium salt hydrate), 2 mM Tris(2-carboxyethyl) phosphine hydrochloride (TCEP), 0.2 mM Sodium orthovanadate, 0.2% (w/v) n-dodecyl β-D-maltoside (DDM), and EDTA-free protease inhibitor (Sigma-Aldrich, USA). Cell suspension was subjected to five times flash-freezing in liquid nitrogen and rapid thawing in water to facilitate cell lysis. After centrifugation at 21,000 g for 20 min at 4 °C, the supernatant was transferred to a new tube and the protein concentration was measured by the BCA assay kit (Thermo Fisher Scientific, USA). Then, samples heated to 37 °C were taken 10 μg of protein, and the same volume of protein was taken at other temperature points to prepare the MS samples.

### Preparation of MS samples
Samples were prepared by SISPROT[58,59], a spin tip-based device, as previously described. Briefly, SISPROT device was fabricated by packing several plugs of C18 disk (3 M Empore, USA) into a standard 200 μL pipette tip and then introducing certain amounts of mixed beads. The mixed beads are composed of POROS SCX beads and POROS SAX beads (Applied Biosystems, USA) in a ratio of 1:1. Subsequent steps of sample loading, protein reduction, alkylation, digestion, TMT pro plexs (Thermo Fisher Scientific, USA) -labeling and desalting were all done on the SISPROT tip. The TMT mixed sample was dried using a vacuum centrifugal evaporator and redissolved with 1% (v/v) formic acid followed by fractionating into six fractions with a stepwise increasing gradient of ACN. Finally, eluted peptide samples were lyophilized to dryness and redissolved in 0.1% (v/v) formic acid in water for nano-LC-MS/MS analysis.

### MS analysis and data Analysis
Each sample was diluted with 0.1% FA prior to separation on 20 cm × 100 μm EASY-Spray C18 LC column with a 135 min gradient on an UltiMate 3000 HPLC system (Thermo Fisher Scientific). The mobile phase was solvent A (0.5% acetic acid in water) and solvent B (80% ACN, 0.5% acetic acid in water). Data were acquired by Orbitrap Exploris 480 mass spectrometer (Thermo Fisher Scientific): MS1 scan resolution was 60,000 and MS/MS scan resolution was 30,000 using turbo TMT. Raw files were searched using Proteome Discoverer (PD) software (Version 2.4, Thermo Fisher Scientific) against the human proteome fasta database (Uniprot, 20376 entries, downloaded on May 03, 2022). The maximum missed cleavage for trypsin digestion was set to 2. The mass tolerance for peptide precursors was 10 ppm and the mass tolerance for fragment ions was 0.02 Da. Carbamidomethyl of cysteine residues, TMT pro modification of lysine residues and TMT pro modification of peptide N-terminal were selected as fixed modifications while oxidation (M) and deamidation (NQ) were selected as variable modifications. FDR control for protein and peptide is 1% at strict level and 5% at relaxed level. All MS analysis were performed with three independent experiments.

### Detection of ATP levels
ATP measurement was determined by CellTiter-Glo Luminescent Cell Viability Assay (Promega, USA). K562 cells under glucose or glucose-free conditions for 0 h, 8 h, 24 h, and 48 h were seeded in a 96-well plate at 10 thousand cells per well. Cells were lysed using Cell Titer-Glo® reagent and mixed for 2 min on an orbital shaker. The plate was incubated for 10 min and analyzed by microplate reader (EnSpire), while cells number were measured by blood counting plates. All ATP measurement were performed with three independent experiments.

### Preparation of WB samples
K562 cells under glucose or glucose-free conditions for 8 h, 24 h, 48 h, and 52 h were harvested. Cells were lysed using RIPA buffer containing a final concentration of 50 mM Tris-HCL pH 8.0, 150 mM NaCl, 1% Triton x-100, protease cocktail, and 1 mM PMSF facilitated by freeze-

thawing five times using liquid nitrogen. After centrifugation at 21,000 g for 20 min at 4 °C, the supernatant was transferred to a new tube and the protein concentration was measured by the BCA assay. Cell lysates were analyzed by western blotting.

## Preparation of Co-immunoprecipitation (Co-IP) samples

K562 cells under glucose or glucose-free conditions for 48 h were harvested. Cells were lysed using RIPA buffer as above, sonicated, and then centrifuged at 21,000 g for 20 min at 4 °C. The supernatant was transferred to a new tube and the protein concentration was measured by the BCA assay. Total 1.5 mg cell lysates were then incubated with 4 μg of primary antibody diluted in 700 μl of PBST Rabbit pAb Control IgG (AC005, Abclonal, 1:100), TAF9B antibody (Proteintech, 28713-1-AP,1:3000) and Emerin antibody (Proteintech, 10351-1-AP,1:5000) in a rotation wheel overnight at 4 °C. Next, samples were incubated with Protein A/G Magnetic Beads (HY-K0202, MCE) for 6 h at 4 °C. To remove the unbound antibody, beads were washed fifth times with 1 ml of PBS. 50 μl of protein loading buffer was added to the beads after which they were boiled at 95 °C for 10 min. Samples were then loaded on an SDS–PAGE gel and further processed for western blotting.

## Western blotting

Twenty micrograms of total protein from WB sample while 15 μl of Co-IP samples were resolved on SDS-polyacrylamide gels and transferred to polyvinylidene fluoride (PVDF) membranes. After blocking in phosphate-buffered saline containing 2.5‰ Tween 20 (PBST) and 5% BSA or skim milk powder for 1.5 h, the membranes were incubated with the indicated primary antibodies overnight at 4 °C. The primary antibodies used in this study were Phospho-Rb (Ser780) (CST, 8180S,1:1000), Phospho-Histone H3(Ser10) (CST, 3377S,1:1000), Phospho-CDK Substrate (CST, 14371S,1:1000), β-Tubulin (CST, 2128S,1:1000), GAPDH (Proteintech, 10494-1-AP,1:5000), MYH9 (Proteintech, 11128-1-AP,1:10,000), Emerin antibody (Proteintech, 10351-1-AP,1:5000), TAF9B antibody (Proteintech, 28713-1-AP,1:3000), TADA3L antibody (Proteintech, 10839-1-AP,1:4500). Then, the membranes were washed 3 × 10 min in TBST. Next, membranes were incubated with secondary antibody for 1 h at room temperature. The following secondary antibody were used: HRP-labeled Goat Anti-Rabbit IgG(H + L) (Beyotime, A0208,1:2000), IPKine HRP, Mouse anti-Rabbit IgG LCS (Abbkine Scientific, A25022,1:2000), IPKine HRP, Goat Anti-Rabbit IgG HCS (Abbkine Scientific, A25222,1:2000). The relative density of each band was analyzed on an Odyssey infrared scanner (LICOR Bioscience, Lincoln, NE, USA). GAPDH and β-Tubulin were used as loading control. All WBs were performed with at least three independent experiments, and quantified with ImageJ software (version ij153-winjava8). Uncropped gel images and replicates were included in the Source Data file.

## MS data normalization

The thermal solubility of protein was obtained by dividing its abundance at different temperatures by its abundance at 37 °C. These readings of every protein from each MS run is derived from the average reading of their isoforms (e.g., PXXXXX-1, PXXXXX-2) weighted according to the number of quantifying PSM (i.e., The number of PSMs used for quantification). In Slim-TPCA, because the data points were not sufficient to fit the logistic curve, we use median normalization to align data. For samples at the same temperature in different treatments/conditions, we assumed that the overall thermal stability of the proteome would not change. Let $M_{x,t} = (m_{x,1,t} \dots m_{x,n,t})$ be the subset of soluble fraction for $n$ proteins under treatment $x$ at the temperature point $t$. For subsets $M_{x,t}$ with the same temperature point $t$ and different treatments $x$, the median $x$ of medians of $n$ proteins of each subset is computed. Subsequently, thermal solubility of all proteins in each subset $M_{x,t}$ are added/subtracted with a fixed value so that all subset have the same median $x$. When using 3 temperatures, we

recommend heating the samples at 37 °C, 49 °C and 58 °C in K562 cells. When using 4 temperatures, we recommend heating the samples at 37 °C, 46 °C, 55 °C and 61 °C in K562 cells.

## MS data filtering

The MS data from the three replicate glucose deprivation experiments was then filtered after normalization. To obtain more robust results, we selected proteins detected in at least 2 replicates and with 3 or more PSMs in total for subsequent analysis. Soluble fractions were obtained by dividing the abundance of proteins at different temperatures by their abundance at 37 °C. Since most proteins would theoretically have a lower soluble fraction after heating than they do at 37 °C, we screened out proteins with solubility fractions higher than 1.2 at 49 °C and 58 °C.

## Empirical statistical assessment of TPCA signature

In the conventional TPCA method, an empirical statistical assessment is used to identify complexes with non-random TPCA signatures. In Slim-TPCA, in order to identify which pairs or complexes exhibit non-random TPCA behavior, we continue the approach adopted in the original TPCA work with a few refinements. For each protein complex, we compute the average Manhattan distance (denote as $D_{avg}$) between melting curves among all pairs of subunits of a protein complex.

Specifically, let $P_A = \{p_1, p_2 \dots p_n\}$ be the set of $n$ unique subunits of protein complex A with solubility data. The average Manhattan distance among all unique subunit pairs ($M_{avg}$) from $P_A$ is computed as

$$D_{avg}(P_A) = \left( \sum_{\substack{1 \le x < n \\ x < y \le n}} d(p_x, p_y) \right) / m \qquad (1)$$

in which $m$, the number of unique subunit pairs from $P_A$, is equal to $(n^2 - n)/2$, and $d(p_x, p_y)$, the Manhattan distance between melting curves of protein $p_x$ and $p_y$, is computed as

$$d(p_x, p_y) = \sum_{i=1}^{t} |x_i - y_i| \qquad (2)$$

where $(x_1, x_2, x_3)$ and $(y_1, y_2, y_3)$ denote the normalized solubility of protein $p_x$ and $p_y$ at $t$ temperature points.

To assess the statistical significance of observed Manhattan distance, we compare $d(p_x, p_y)$ with Manhattan distance of 10,000 randomly selected protein pairs or protein sets of size n. The TPCA Signature (P-value) of observed Manhattan distance is then estimated as frequency of 10,000 distance $<d(p_x, p_y)$. The protein pairs or complexes with TPCA Signature P-value < 0.05 are defined with pairs or complexes with TPCA signature, meaning that the thermal melting curves of their subunits are close together, in accordance with the TPCA theory.

## Empirical statistical assessment of TPCA Modulation Signature

Here, we have optimized the algorithm for identifying the dynamic modulation by calculating the changes in distance. If $P_A = \{p_1, p_2, \dots, p_n\}$ is the entire set of subunits from complex A with solubility data from both Control and Treatment samples, the absolute and relative difference in distance are computed as

$$\Delta_{abs(Ctrl-Treat)} = M_{avg}(P_{A(Ctrl)}) - M_{avg}(P_{A(Treat)}) \qquad (3)$$

$$\Delta_{rel(Ctrl-Treat)} = \frac{M_{avg}(P_{A(Ctrl)}) - M_{avg}(P_{A(Treat)})}{M_{avg}(P_{A(Ctrl)})} \qquad (4)$$

where $M_{avg}(P_{A(Ctrl)})$ and $M_{avg}(P_{A(Treat)})$ is computed from solubility data derived from Control and Treatment samples, respectively.

To assess the statistical significance of observed $\Delta_{(Ctrl-Treat)}$, we compute $\Delta$ for 10,000 randomly selected protein set of size $n$. The TPCA Modulation Signature ($P$-value) of observed $\Delta_{(Ctrl-Treat)}$ is then estimated as frequency of random protein set that result in $\Delta < \Delta_{(Ctrl-Treat)}$. The complexes with TPCA Modulation Signature $P$-values less than 0.05 are identified as convergent complexes, and those with TPCA Modulation Signature $P$-values greater than 0.95 as divergent complexes.

**Empirical statistical assessment of TPCA Modulation Z-score**

In addition to using the TPCA modulation signature to identify complexes with significant dynamic changes, we also borrow the TPCA modulation z-score from previous paper to detect complex dynamics after virus infection. We used $Mx_{c,i}$, where $Mx_{c,i} = \frac{1}{1+M_{c,i}}$ and $M_{c,i}$ is the average Manhattan distance of proteins in complex (c) at specific infection time point (i; 24, 48, 72, and 96 hpi or mock), calculated from 10, 4, or 3 temperature points. Z-scores are then calculated based on the null-distribution of $Mx_{c,i}$ generated from random protein complexes with same size of proteins as in complex (c) with 10,000 iterations.

**Beta distribution fitting algorithm**

In empirical statistical assessment of TPCA signature and TPCA modulation signature, 10,000 random complexes of size $n$ are sampled and their average distances or changes in average distances are calculated. To save computational time for high throughput data, here we use a smaller sampling size in conjunction with Beta distribution fitting instead of an excessive sampling. We choose to sample 500 random complexes, calculate their average distance or change in average distance, and then fit them with the Beta distribution as

$$f(x, a, b) = \frac{\Gamma(a+b)x^{a-1}(1-x)^{b-1}}{\Gamma(a)\Gamma(b)} \tag{5}$$

where $\Gamma$ is the gamma function. The TPCA signature ($p$-value) and TPCA modulation signature are defined as the probability of having a more extreme value in the probability density function of the fitted Beta distribution than the value of the putative complex.

**Reporting summary**

Further information on research design is available in the Nature Portfolio Reporting Summary linked to this article.

# Data availability

All the raw MS data have been deposited to the ProteomeXchange Consortium via the iProX partner repository[60,61] with the dataset identifier PXD040078. Source data are provided with this paper.

# Code availability

The installation package of Slim-TPCA can be downloaded directly from PyPI (https://pypi.org/project/Slim-TPCA/). Package documentation can be acquired online via https://slim-tpca.readthedocs.io/en/latest/index.html. Testing data can be downloaded from GitHub (https://github.com/wangjun258/Slim_TPCA_examples)[62].

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

## Acknowledgements

This work is supported by grants from and National Key Research and Development Program of China (No. 2021YFA1302603), Shenzhen Innovation of Science and Technology Commission (No. JCY20200109140814408) and the National Natural Science Foundation of China (Nos. 22074060, 22150610470) awarded to C.S.H.T. Figure 5a was created with MedPeer (www.medpeer.cn).

## Author contributions

C.S.H.T. conceived and led the project. S.S. analyzed data and wrote the algorithms. Z.Z., F.L., K.L. and S.Z. performed the biological experiments. J.W. and A.H. inspect and organized the code. J.-H.L. guided the biological experiments while R.T. guided the TMT-based proteomics experiment. S.S., Z.Z. and C.S.H.T. wrote the paper.

## Competing interests

The authors declare no competing interests.
