## [Peer Review File · Nature Communications]

Improved in situ Characterization of Protein Complex Dynamics at Scale with Thermal Proximity Co-AggregationREVIEWER COMMENTS

Reviewer #1 (Remarks to the Author):

Sun et al. present an extension of the TPCA method for in situ detection of protein-protein interactions (PPIs) and protein complex assembly. They explore different metrics for comparing the thermal profile of different proteins, find a superior one for the analysis than the previously used one and present a slimmer experimental format which allows greater throughput at a small cost of reduced accuracy. Moreover they show how this format can be used to study dynamically modulated complexes and present an application to time course data of glucose depletion in K562 cells. Due to the combination of the presented technological advances together with an useful software package I believe that the presented work offers value for the community, but I feel it would strongly benefit from the authors addressing several points:

Major comments:

- The concept of multiplexing different treatment conditions within the same TMT set to achieve higher accuracy is not novel for the application of TPP/TPCA and was first introduced by Becher et al., 2016, Nat. Chem. Biol. and then in combination with sample/channel usage reduction in a similar way as presented here by a method termed PISA (Gaetani et al., 2019, J. Proteome Res.) and later for classical TPP with use of TMT16 by Zinn et al., 2021, J. Proteome Res.. The authors should acknowledge this and give proper credit to prior work when introducing their approach.
- The reduction of sensitivity for detection of PPIs with the subsampled format is only presented as a reduction in ROC AUC. Could the authors analyze which PPIs are no longer detected, or vice versa which false positive interactions are now predicted? Is there a pattern in these groups, e.g. lower than average melting point or high molecular weight? This would be important to understand whether the modification of the original method introduces any biases.
- An inherent limitation of the approach is the dependence on external database/annotation source of PPIs, could the authors discuss this also in the context of using the method for making de novo predictions, e.g. in less well studied species?
- I have tried to install their Python package and it needs several other packages as dependencies. It would be nice if this was better documented or that the installation of dependencies was handled automatically. Moreover, the example code uses data which is not accessible for external users. It would thus be nice if that would be changed (could be also simulated data) so that potential users can easily understand what the input data formats need to be and can easily try out the implemented methods on readily available data.

Minor comments

The article features several language/grammar errors, some examples below:

- Abstract middle part „On the other hand „ instead of „On other hand“
- Last sentence abstract „various algorithmic improvements [...] are“ instead of „is“
- p.8 first word should be „strengthened“

Reviewer #2 (Remarks to the Author):

Key results

The authors describe a workflow for identifying potential protein interactions based on the shape of their respective melt profiles. This workflow is based on their previously reported TPCA with the following modifications. First, the proposed workflow uses a smaller number of temperatures in the heating gradient. Second, the analysis uses a Manhattan distance rather than a Euclidean distance to describe the protein melt curve shapes. Third, their analysis uses relative distance instead of absolute distance. Fourth, they have developed a p-value calculation workflow that is less time intensive by

using a Beta distribution rather than a bootstrapping approach. Their new data analysis pipeline has been published as a publicly available python program and was validated using a novel temporal data set.

Validity

The authors have attempted to account for biological variability by including experiments that are in triplicate. This is of great benefit to assessing the validity of the presented data. The authors also do a really nice job of testing their new workflow using various publicly available data sets including their temporal experiment. The discussion of how glucose deprivation reflects the results they observe particularly around mitochondrial axes is very well thought out.

It is unclear whether the simplification of the temperature gradient is only applicable for the mammalian cell lines used in this experiment workflow or whether other cell lines with lower/higher average melts would be adaptable to this modified workflow as well. The trade-offs of using the modified workflow vs. the original 10 temperature workflow are also not apparent from the report. There is an obvious benefit from the increased throughput but it is not clear what is lost by simplifying from 10 temperatures to 3 temperatures. The authors use Pearson correlations to show that the values remain high from 10 temperatures to 3 temperatures but it is unclear whether this is an accurate representation of correlation considering that these data sets are most likely not normally distributed.

Significance

This work and modified workflow significantly increase the throughput of a TPCA experiment and consequently decrease the level of noise encountered when analyzing samples in multiple MS experiments. The benefit of the modified workflow is even more significant when considering experiments where multiple drug treatments and time points are incorporated. The Slim-TPCA approach has great potential to advance the field and improve the quality of findings from TPCA experiments.

Data and methodology

The quality of data appears to be very good based on the results observed in the bioinformatic evaluation that is done throughout. At the same time, it is unclear whether the quality of the results from the new workflow would hold up in a discovery based experiment where annotations are not leveraged. Would novel protein relationships be detected in an output where there is now more noise (since 10 temperatures are no longer used)?

Evaluation of Data/Methodology Quality has also been documented on a Figure basis below:

Figure 1c: While the analysis including whole cells and cell lysates is interesting, it does not align with the rest of this section. At this point in the narrative by the authors, the choice of distance measures is being chosen. It is unclear how the choice of sample preparation (i.e. lysis vs. whole cell) is being used to decide on the distance algorithm.

Figure 1d: Consider adding an axis break so that the values for the Cosine and Pearson can be seen. It should also be considered that the error bars for each are wide enough that they overlap with each other across pretty much all temperatures, including those at 10 temperatures. It is also unclear which bars at 10 temperatures correspond to the different methods as the colors are not clearly visible.

Figure 2: It is not clear what the criteria was for selecting 3 or 4 temperatures as acceptable for approximating the full 10 temperature experiment. It is also not clear what the trade-off (impact to false positives) is by only using 3 or 4 temperatures. As was stated previously, there is a decrease in AUC that appears to be minimal but there is also a large impact to error bars around each median

AUC.

Figure 2d. The definition of the Dynamic p-value is not clear based on what is described in the methods section. Is this purely a p-value calculated in a temporal experiment? It is unclear how this is calculated or how it compares/differs from 2c. Related publications (i.e. Tan CSH, Go KD, Bisteau X, Dai L, Yong CH, Prabhu N, Ozturk MB, Lim YT, Sreekumar L, Lenggqvist J, Tergaonkar V, Kaldis P, Sobota RM, Nordlund P. Thermal proximity coaggregation for system-wide profiling of protein complex dynamics in cells. *Science*. 2018 Mar 9;359(6380):1170-1177. doi: 10.1126/science.aan0346. Epub 2018 Feb 8. PMID: 29439025.) was also consulted and it was not defined here.

Figure 2c through 2e: The term "Signature" is used to describe the values in these plots. Is the "Signature" different than the Euclidean distance? Is it the p-value for the Euclidean distance? If not, it would be beneficial and more clear to use the same terminology throughout the manuscript to avoid confusion. The term "signature" is a bit qualitative to this reviewer and does not come across as a quantitative evaluation of melt.

Figure 2e. The meaning of this figure is not clear. It could be helpful to have more labeling of axes. What values are depicted by red vs blue; are these z-scores? It is not clear how the conclusion is made that 3 and 4 temperature results are comparable with the panel on the left of 2e. Is the similarity between the three concluded based on the fact that changes in colors are similar between the three sub-panels? There may be other ways to visualize the data so that it can be reinforced that 3 and 4 temperatures are comparable with the 10 temperature conclusions. It is also not clear what is meant by "matched as close as possible".

Figure 3e. Figure 4d and 4e: The y-axis indicates that the values are percentages. Is this a typo? If these are actually percentages (and not fractions), the values seem to be extremely low.

Figure 3f and 3g. It is difficult to tell which curves correspond to MTX treatment and which are DMSO. The labels on 3g are also missing.

Supplementary Figure S4: It is not clear what the difference is between the three tables listed.

Figure 4g and 4h: Are the complexes reported by the Beta method false positives? While it is initially encouraging that the method with lower computational expense reports all complexes from the sample method, it would be good to know how many are false positives.

Figure 5: In plots like this and those in the supplementary section, it would be beneficial if error bars were present on each of the points so that the variability from the replicate experiments can be evaluated. Error bars would allow the user to evaluate whether the conditions that are assumed from changes in curve shape are actually within the "noise" of the system.

Supplementary Fig.14 and Supplementary Fig.15 are referenced in the text but could not be located in the supplementary material.

Analytical approach

Overall, the approaches taken to analyze data and arrive at the improved workflow is well thought out and fairly easy to follow. Some of the techniques used to assess the quality of the new workflow (and their inherent assumptions) are a little difficult to understand (i.e. Beta Distribution) but overall the flow of analysis is very good.

The conclusion that the modified workflow has acceptable results when compared to the 10 temperature workflow is based on use of Pearson correlation. It is not clear that this is an acceptable measurement due to the fact that the data sets being compared are not normally distributed but are

are skewed.

Assessment of some data could not be done due to some axes and legends not being fully complete.

Since the Manhattan distance is chosen over the Euclidean distance, it might be beneficial for the authors to speculate why it is in principle a better approximation and whether this only applies for 3-4 temperature data sets (and not the 10 temperature set).

Evaluation of Analytical Approach has also been documented on a Figure basis below:

Figure 1b: This figure is not currently referenced in the main text. It is unclear how the error bars on the AUC for 2 temperatures are smaller in magnitude than the other higher temperatures in the figure. It is expected that with a smaller number of data points, there would be a larger amount of variability around the mean. Perhaps consider including the standard deviation as a variable when determining the optimal number of temperatures. This is relevant since it is being argued that there is a large decrease in AUC with the lower number of temperatures. It is also important to note that while the magnitude in difference between the high and low number of temperatures is only 0.06 (0.71 and 0.65), this is a large number when considering analysis of proteomic data sets where number of proteins can approach 10,000. This is also relevant in Supplementary Figure 1 where described improvements in AUC between "Cityblock" and "Euclidean" distance measurements are on the order of 0.005.

Figure 1d: It is concluded in this first section that the Manhattan analysis is better than the other methods based on the higher median value. It is not discussed, however, that the error bars for each of these data sets are extremely large and that there are some methods that actually have smaller error bars (i.e. Chebyshev). Consider explaining what criteria were used for selecting Manhattan as compared to Euclidean. It may also be prudent in the discussion section to explain why the chosen method better approximates (based on the details of the calculation).

Figure 2c and 2d: The data for each axis does not appear to be normally distributed but instead appears to be skewed. Is the Pearson coefficient the correct value to describe the correlation between the two axes or would something like the Spearman be more suitable for the non-normally distributed data? This is particularly important due to the fact that a large amount of the data skews at the higher values (presumably those with more significant p-values). For example, there are many points that would be below a cut-off of 2 on one axis but well above 2 for the other axis. See Schober, P. , Boer, C. & Schwarte, L. (2018). Correlation Coefficients: Appropriate Use and Interpretation. *Anesthesia & Analgesia*, 126 (5), 1763-1768. doi: 10.1213/ANE.0000000000002864.

Figure 3a and 3b. The use of a Pearson correlation is most likely not suitable due to the fact that both axes are not normally distributed.

In the second paragraph of the last section prior to discussion, the following statement is made: "Three sets of biological replicates are performed, and the data collected are pooled for integrative analysis to maximize signal and reduce stochastic variation." It is not clear what this statement means and whether the variability observed from the replicates was truly utilized in the analysis.

Suggested improvements

The authors use the K562 cell line for the identification of optimal temperatures for the simplified version of their workflow. The authors do use a data set collected using a fibroblast cell line as a comparator. It would be beneficial, however, for the optimal temperatures to be compared using a different species (i.e. yeast) where the mean melt temperature could be higher or lower than that of K562. This would not necessarily involve generation of a novel data set but could be done using a data

mining approach. This is particularly important if this workflow is recommended for use beyond that of mammalian systems alone.

It would be beneficial for the authors to describe the sample heating procedure that was used as it is integral to the experiments described. It is also stated at the end of the cell culture experiment that a Bradford assay was conducted; was the protein concentration adjusted prior to heat treatment?

While the authors clearly state the benefits of using the modified workflow (i.e. time and sample throughput), they do not discuss the deficiencies when compared to the original 10 temperature method. Possibly another way to state would be whether the authors are arguing that the original method using Euclidean distance is no longer a valid approach for measuring protein association.

Clarity and context

The authors provide a discussion of their work in a way that is fairly accessible to scientists of various background. The order of steps taken by the authors to arrive at their final new workflow is very well explained. Some of the statistical methods and approaches used by the authors (i.e. Pearsons) may not be the best for these data considering that the data are not normally distributed.

References

The principle of drug-protein interaction affecting protein melt temperature is actually a bit more established than that suggested by the references listed by the authors. It would be helpful to list older references such as:

Lo, M. C., Aulabaugh, A., Jin, G., Cowling, R., Bard, J., Malamas, M., & Ellestad, G. (2004). Evaluation of fluorescence-based thermal shift assays for hit identification in drug discovery. *Anal Biochem*, 332(1), 153-159. doi:10.1016/j.ab.2004.04.031

Cimpmperman, P., Baranauskiene, L., Jachimoviciute, S., Jachno, J., Torresan, J., Michailoviene, V., . . . Matulis, D. (2008). A quantitative model of thermal stabilization and destabilization of proteins by ligands. *Biophys J*, 95(7), 3222-3231. doi:10.1529/biophysj.108.134973

Your expertise

Unable to assess the following: Quality of the raw MS data. Quality of the Python based code. FACS gating strategy.

Reviewer #3 (Remarks to the Author):

Protein thermal stability can be affected by protein-protein interaction, protein metabolite interaction, protein posttranslational modification, protein localization, etc. Mass spec-based proteomics supports measuring protein thermal stability changes in a high-throughput manner. The group previously published proteomics-based TPCA method for profiling protein complexes, under the assumption that proteins of similar melting behaviors belong to the same complex. Sun et al. described a simplified TPCA method via simulation in this manuscript, reducing the number of experiment temperatures from 10 to 3 or 4, improving the efficiency of the assay. The authors further used a glucose deprivation experiment to benchmark the simplified TPCA method. The manuscript is relatively well written, but the significance of the simplified TPCA method is limited and the showcase experiment is also weak.

Major comments:

1. The key assumption of TPCA method is that proteins of similar melting behaviors belong to the same complex, and the melting behavior shift of a protein complex member indicates the protein leaves or joins the complex. This is enabled by the relatively high number of experimental

temperatures thus the high resolution of melting behaviors, on condition that the quantitative accuracy achieved in the mass spec-based proteomics assay is also high. However, protein-protein interaction is not the only factor that can affect protein thermal stability and a high resolution (aka, a high number of temperature and high quantitative accuracy) is needed in order to make convincing conclusions. In this manuscript, the authors reduced the number of temperatures to 3 or 4, compromising the resolution and the robustness of the previous well-established data normalization method. It is true that the assay efficiency is improved by reducing the number of temperatures, but the confidence in the conclusion is also compromised to some extent. The author did carry out a series of simulation based on the previous TPCA publication and another study to justify the reduced number of temperatures and selection of distance calculation method, but the transferability of the conclusion needs to be future demonstrated considering the potential over-fitting risk. Besides, the measured protein melting behavior is also affected by other experimental conditions, including lysis buffer composition, in cell vs lysate. Thus, the emphasis on the optimal three or four temperatures may mislead the community.

2. Considering the limited number of temperatures used in the showcase experiment (glucose deprivation assay), the conclusion more of demonstrated the protein thermal stability changes after glucose deprivation. However, the thermal stability changes may not necessarily indicate protein-protein interaction changes, it could be PTM changes, protein localization changes, etc. It is a stretch to say "Proteome-wide Protein Complex Dynamics" in this case. It may work for some essential, large and abundant protein complexes, but the robustness for proteome-wide protein complexes is still unclear.

3. Some summary statistics should be included in Fig5, for example, how many proteins are quantified, cv, how may protein complexes are investigated in total.

4. Is protein level FDR also controlled? Method only says 1% PSM and peptide FDR.

5. The effect size in many plots (e.g. Fig 5d and some supplemental figures) is not big. Some seems to fall in the range of measuring variance of mass spec-based proteomics. Were the three replicates carried out completely independently (from cell culture to data acquisition)? Adding error bars or confidence intervals would help.

6. Supplementary Figure 5, there is really no difference between the two distance calculation methods looking at the plots.

Minor comments:

1. The method section has a paragraph describing "Treatment with Pharmacological Inhibitors". But no main text/data/figure is related to this part of the method.

2. The main text refers to Supplemental Figure 13, 14 and 15, but the figures are not available in the submission.

3. The experimental details for the glucose deprivation assay (Fig 6) are missing in the method section.

4. Figure legends for Supplemental Figure 5-12 are missing. Difficult to figure out what each panel indicates. It's better to label each panel indicating whether it's a convergent or divergent case.

Reviewer #1:

Comment 1 The concept of multiplexing different treatment conditions within the same TMT set to achieve higher accuracy is not novel for the application of TPP/TPCA and was first introduced by Becher et al., 2016, Nat. Chem. Biol. and then in combination with sample/channel usage reduction in a similar way as presented here by a method termed PISA (Gaetani et al., 2019, J. Proteome Res.) and later for classical TPP with use of TMT16 by Zinn et al., 2021, J. Proteome Res. The authors should acknowledge this and give proper credit to prior work when introducing their approach.

Response: Thank you for your feedback. We have revised the manuscript by adding references to the articles mentioned above in the second last paragraph of the introduction section, as shown in the text below:

“Similar ideas have been shown to be used to improve the TPP/MS-CETSA method. For example, the PISA³⁶ and iTSA³⁷ methods analyze different treatment conditions within the same TMT set for target deconvolution^{38,39}.”

Comment 2 The reduction of sensitivity for detection of PPIs with the subsampled format is only presented as a reduction in ROC AUC. Could the authors analyze which PPIs are no longer detected, or vice versa which false positive interactions are now predicted? Is there a pattern in these groups, e.g. lower than average melting point or high molecular weight? This would be important to understand whether the modification of the original method introduces any biases.

Response: Thank you for your thoughtful feedback as we concur there could be trade-offs when using three or four temperature points. To gain insight into the analysis suggested by the reviewer, we first need to define a cutoff to identify a set of true positives. In this work, we had performed statistical evaluation to assess the stochastic likelihood (i.e. p-value) of any Manhattan distance observed. First, we observed the p-values of TPCA signature for protein pairs obtained with 10 compared to 3 or 4 optimal temperatures identified are not significantly different (differences in p-values i.e. Δp mostly near 0, see figures below) and are very well correlated (average Pearson's and Spearman R of 0.90 and 0.95 respectively in the intact cell data, and 0.92 and 0.96 respectively in the cell lysate data) reflecting directionality is preserved with overall only slight variation in p-values (see figures below). Nevertheless, we define the set of PPI with very strong TPCA signal under 10 temperatures (defined as those with p-value < 0.01) as true positives (which can be viewed as the subset of known PPI with potentially high recurring rate), and its subset that do not possess TPCA signal (defined as those with p-value > 0.1) under 3 or 4 temperatures as false negatives. Conversely, we define false positives as PPI with non-existent TPCA signal under 10 temperatures (p-value > 0.1) but exhibit strong TPCA signal under 3 or 4 temperatures (p-value < 0.01).

Legend: Difference in p -values computed for protein pairs using optimal subset of temperatures identified, and the full set of temperatures. (a-b) The symmetric density plots illustrates that the TPCA signature p -values of most protein pairs did not change significantly in intact cells and cell lysates respectively between the full temperature set and identified subset.

Legend: Scatter plot of TPCA signatures (p -values) for random protein pairs. Correlation of TPCA signatures (p -values) of protein pairs from 10 temperature points and identified subset. We observed the TPCA signatures (p -values) of protein pairs obtained with 10 and 3 or 4 temperatures are not significantly different but are well correlated (average Pearson's and Spearman R of 0.90 (a) and 0.95 (b) respectively in the intact cell data and 0.92 (c) and 0.96 (d) respectively in the cell lysate data).

In summary, we define false negative pairs as:

$$(pvalue_{10 \text{ temp}} < 0.01) \text{ AND } [(pvalue_{3 \text{ temp}} > 0.1) \text{ OR } (pvalue_{4 \text{ temp}} > 0.1)]$$

And false positive pairs as:

$$(pvalue_{10 \text{ temp}} > 0.1) \text{ AND } [(pvalue_{3 \text{ temp}} < 0.01) \text{ OR } (pvalue_{4 \text{ temp}} < 0.01)]$$

From the intact cell data, 582 PPI have very strong TPCA signal (p-value <0.01) under 10 temperatures but 2 of them (0.34%) are without TPCA signal (p-value > 0.1) under 3 or 4 temperatures. A total of 3132 PPI have no TPCA signal (p-value > 0.1) under 10 temperatures, and only 7 (0.22%) of them exhibit TPCA signal under 3 or 4 temperatures (p-value < 0.01). For the cell lysate data, there are 607 PPI with TPCA signal, of which none (0%) are found to be false-negative under 3 or 4 temperature points. Of the 3675 PPI without TPCA signal under 10 temperatures, 22 protein pairs (0.60%) exhibit TPCA signal under 3 or 4 temperatures. Thus, using the 3 or 4 temperatures identified do not affect overall analysis of PPI with very few false positives and false negatives.

To detect possible trend, we compared the molecular weights and melting points of proteins from false-positive or false-negative pairs to all proteins in the data (see figures below). The molecular weights of these proteins were not significantly different from that of the whole proteome (t-tests), though the melting points of the false-positive proteins in the intact cell data were significantly lower from that of the whole proteome. However, the absolute value of the difference in the melting points was small, so the significance may be due to the small number of false-positive proteins.

Legend: Box plots of molecular weight and melting temperature of proteins in all protein pairs compared to proteins in false-positive and false-negative pairs for *intact cell*. T-tests were used to assess significant differences.

Legend: Box plots of molecular weight and melting temperature of proteins in all protein pairs compared to proteins in false-positive and false-negative pairs for **cell lysate**. T-tests were used to assess significant differences.

Next, we examined protein complexes dynamically modulated during MTX treatment. Overall, the TPCA modulation signatures of the complexes computed with 3 or 4 temperatures correlate well with the results obtained using 10 temperatures (Fig. 2d, main text). Nevertheless, of the 48 complexes identified as dynamically modulated originally (TPCA modulation p-value < 0.05), 4 complexes (8.3%) no longer exhibit TPCA signal (TPCA modulation p-value > 0.1) with 3 or 4 temperatures. In contrast, of the 705 complexes not identified as modulated under 10 temperatures (TPCA modulation p-value > 0.1), 4 complexes (0.57%) changed significantly under 3 or 4 temperature instead (TPCA modulation p-value < 0.05) and were deemed as false positives. Overall, the difference in number of modulated protein complexes is small.

Intuitively, Slim-TPC, with less data points (temperatures), will involve some trade-offs although it facilitates more flexible experimental design and reduce variations from multiple MS run (batch effect). However, after comparing the results of protein pairs and complexes obtained by the traditional method with those obtained by Slim-TPCA analysis, we believe that the impact of these trade-offs is relatively small and acceptable, particularly when considering the elimination of batch variation arising from multiple MS runs. For the modulated protein complexes, the original analysis was performed comparing data obtained from multiple batches of TMT experiments which undoubtedly increased stochastic variation and hence potentially more false positives (modulated protein complexes). Slim-TPCA, with reduced number of temperatures, permits multiple conditions to be analyzed concurrently on MS, thus eliminate stochastic variation between different batches of MS run. Nevertheless, there is still possibility of false positives and negatives which we have highlighted in the revised manuscript, and recommend that the community conduct an assessment before selecting a method and to determine whether Slim-TPCA or traditional TPCA is an appropriate method for the research topic.

Accordingly, we have added the following in our main text.

“In summary, we have incorporated multiple improvements for TPCA analysis of protein complex dynamics that include new statistical model and using a subset of temperatures. The latter permits multiple conditions and replicates to be analyzed concurrently on MS instrument using multiplexing reagents. This eliminates batch variation across multiple MS runs in the traditional TPCA analysis and reduce false positives. Nevertheless, the optimization using reduced number of temperatures could also involve trade-offs, for example, protein interactions that can be identified by traditional TPCA methods may be missed in Slim-TPCA, or vice versa, Slim-TPCA may identify false-positive protein interactions.”

“For other species, we recommend to first perform conventional TPCA experiments to determine the melting temperature and to verify the feasibility of the Slim-TPCA method in this species sample”

Comment 3 An inherent limitation of the approach is the dependence on external database/annotation source of PPIs, could the authors discuss this also in the context of using the method for making de novo predictions, e.g. in less well studied species?

Response: Thank you for your interest on the possibility of extending TPCA for predicting de novo protein-protein interactions. While we hope TPCA could be used to make de novo protein-protein interactions, particularly for less-studied species lacking much data on protein interactions and complexes, we felt the data obtained is insufficient by itself for making prediction with accuracy acceptable for practical use. Allow us to illustrate this with figure below (from Tan et al, 2018) showing the melting curves of about 6000 human proteins extrapolated from data obtained with 10 temperatures. The red lines are the 4 subunits of Condensin I complex. In between the two outermost red lines lie the melting curves for about 300 proteins. Assuming we already know the proteins represented by two outmost red lines form a complex, and there are another two unknown subunits in between the two outmost red lines, the accuracy of using melting curve to predict the other two subunits will be 1 in 150.

Legend: Protein melting curves of selected protein complexes: red lines, complex subunits. For the leftmost plot, the solubility curves of ~6000 other proteins are plotted in gray.

Succinctly, the inherent challenge in using the approach to make de novo predictions is there are too many non-interacting protein pairs with very similar melting curves while interacting protein pairs can have significant but are not necessary the highest similarity in melting curves. Intuitively, this is because many interactions do not occur at maximum frequency or occurrence (i.e. every protein is in a complex) such as for signalling complexes. Thus, based on the understanding that the interaction of two proteins will not occur at max occurrence in cells (hence will not have most similar melting curves), we acknowledge the limitations of the TPCA method for predicting protein interactions with accuracy that will be of practical use. However, the method's capability to generate relevant predictions in specific scenarios, such as viral-host interactions, highlights its potential and practical applicability in certain contexts (Hashimoto et al., 2020) or for some proteins, but more work is needed to establish generality of the approach.

As the frequency of interactions could change across cellular states which are reflected in change in similarity of melting curves, we have advocated using TPCA to monitor the assembly state of known protein complexes across cellular states and conditions instead. For less well studied species lacking PPI data, we think existing predicted protein-protein interaction (PPI) prediction tools could be used in conjunction with TPCA to improve accuracy. Indeed, existing PPI prediction methods that leverage various information sources, including predicted 3D structures, have demonstrated remarkable results on benchmark datasets (Zhang et al., 2012). Notably, the recent advancements in PPI prediction and the integration of 3D structural information, have propelled the precision levels to an impressive 85% or above (Baranwal et al., 2022). Nonetheless, these methods at best suggest the possibility of interaction between two proteins. By overlaying these predicted PPI with TPCA analysis of data from whole cell might reveal subset of PPI that occurred or occurred at higher frequency in certain conditions.

Once again, we genuinely appreciate the thorough evaluation of our work provided by the esteemed reviewer. We assure you that we hold your feedback and suggestion in high regard and will carefully consider your suggestion for further refinement.

Reference

- Baranwal, M., Magner, A., Saldinger, J., Turali-Emre, E.S., Elvati, P., Kozarekar, S., VanEpps, J.S., Kotov, N.A., Violi, A., Hero, A.O., 2022. *Struct2Graph: a graph attention network for structure based predictions of protein–protein interactions*. *BMC Bioinformatics* 23, 370. <https://doi.org/10.1186/s12859-022-04910-9>
- Hashimoto, Y., Sheng, X., Murray-Nerger, L.A., Cristea, I.M., 2020. *Temporal dynamics of protein complex formation and dissociation during human cytomegalovirus infection*. *Nat Commun* 11, 806. <https://doi.org/10.1038/s41467-020-14586-5>
- Tan, C.S.H., Go, K.D., Bisteau, X., Dai, L., Yong, C.H., Prabhu, N., Ozturk, M.B., Lim, Y.T., Sreekumar, L., Lengqvist, J., Tergaonkar, V., Kaldis, P., Sobota, R.M., Nordlund, P., 2018. *Thermal proximity coaggregation for system-wide profiling of protein complex dynamics in cells*. *Science* 359, 1170–1177. <https://doi.org/10.1126/science.aan0346>
- Zhang, Q.C., Petrey, D., Garzón, J.I., Deng, L., Honig, B., 2012. *PrePPI: a structure-informed database of protein–protein interactions*. *Nucleic Acids Research* 41, D828–D833. <https://doi.org/10.1093/nar/gks1231>

Comment 4 I have tried to install their Python package and it needs several other packages as dependencies. It would be nice if this was better documented or that the installation of dependencies was handled automatically. Moreover, the example code uses data which is not accessible for external users. It would thus be nice if that would be changed (could be also simulated data) so that potential users can easily understand what the input data formats need to be and can easily try out the implemented methods on readily available data.

Response: Thank you for your feedback. We have incorporated your suggestions and made necessary changes to the package. The updated package includes all dependent packages, which will be automatically loaded during the initial installation process. Furthermore, we have included sample data in the GitHub repository (https://github.com/wangjun258/Slim_TPCA_examples) and modified the sample code to directly refer to the sample data in the test_data directory in the GitHub repository directly. This allows users to simply clone the repository and run the sample data from their current directory without any additional setup.

Comment 5 The article features several language/grammar errors, some examples below:

- Abstract middle part „On the other hand „, instead of „On other hand“
- Last sentence abstract „various algorithmic improvements [...] are“ instead of „is“
- p.8 first word should be „strengthened“

Response: We apologize for the language/grammar errors. In this revision, we tried our

best to reduce grammatical errors. The point-to-point response to the issues are listed as follow.

Comment 6 Abstract middle part „On the other hand „, instead of „On other hand“. Last sentence abstract „various algorithmic improvements [...] are“ instead of „is“

Response: In the revised manuscript, the grammatical errors of abstract have been corrected.

“On the other hand, protein complexes involved in protein transport and degradation are found increasingly associated revealing their involvement in metabolic reprogramming during glucose deprivation. In summary. Slim-TPCA is an efficient strategy for proteome-wide characterization of protein complexes. The various algorithmic improvement of Slim-TPC are available as Python package at <https://pypi.org/project/Slim-TPCA/>”

Comment 7 p.8 first word should be “strengthened”

Response: We have fixed this and proofread the revised manuscript again.

Reviewer #2:

Comment 8 It is unclear whether the simplification of the temperature gradient is only applicable for the mammalian cell lines used in this experiment workflow or whether other cell lines with lower/higher average melts would be adaptable to this modified workflow as well.

Response: Thank you for your feedback. In the manuscript, we provide suggestions for the selection of temperature points in mammalian cells. Specifically, we suggest picking the intermediate, widely separated temperature points that are evenly distributed around average melting temperature (T_m) will most likely maximally encapsulated TPCA signal embedded across the temperature range tested. However, we did not test this hypothesis using proteomic data from other species. In the revision, we analyzed the proteomics data of *Arabidopsis thaliana* as well as TPP data of 15 other species housed in Meltome database and obtained similar conclusion. The addition in the manuscript are shown below

“Importantly, the identified optimal combination of temperature points is in line with our expectation that picking the intermediate, widely separated temperature points that are evenly distributed around average melting temperature (T_m) maximally encapsulated TPCA signal embedded across temperature range tested. To test this theory, we chose TPP/MS-CETSA data from *Arabidopsis thaliana*, whose proteome has a lower melting temperature than the mammalian proteome. Similarly, the combination of temperatures producing data that correlate most with that from 10 temperature data (Supplementary Fig.6a) are also intermediate, widely, and equally distributed around the average melting temperature of the proteome of *Arabidopsis thaliana* ($\sim 46.6^\circ\text{C}$). Furthermore, we also analyzed datasets from 15 species housed in the Meltome database that include *B. subtilis* and *C. elegans* etc. The data for these species similarly demonstrate that the right combination of temperature points in the Slim-TPCA method can encapsulated information embedded in data from 10 temperatures well (Supplementary Fig.6b-6p). These results suggest the framework proposed in this work could also be adopted for these species. For other species, we recommend to first perform conventional TPP experiments to determine the melting temperature and to verify the feasibility of the Slim-TPCA method in this species sample.”

(Supplementary Fig.6, shown below)

Supplementary Figure 6. Heatmap of correlation between Manhattan distances of 100,000 random protein pairs computed with 3 and 10 temperature points in different species. Data were obtained from studies on *A. thaliana* and from the Meltome database. These results suggest the framework proposed in this work could also be adopted for other species.. (a) *A. thaliana*. (b) *B. subtilis*. (c) *C. elegans*. (d) *D. melanogaster*. (e) *D. rerio*. (f) *E. coli_cells*. (g) *E. coli_lystate*. (h) *Jurkat*. (i) *K562*. (j) *M. musculus_BMDC*. (k) *M. musculus_liver*. (l) *O. antarctica*. (m) *P. torridus*. (n) *S. cerevisiae*. (o) *T. thermophilus_cells*. (p) *T. thermophilus_lystate*.

Comment 9 The trade-offs of using the modified workflow vs. the original 10 temperature workflow are also not apparent from the report. There is an obvious benefit from the increased throughput but it is not clear what is lost by simplifying from 10 temperatures to 3 temperatures.

Response: Thank you for your thoughtful feedback as we concur there could be trade-offs when using three or four temperature points. Similar concern (Comment 2) was also raised by the first reviewer. For your convenience, we reproduced part of our response to the comment below:

In this work, we had performed statistical evaluation to assess the stochastic likelihood (i.e. p-value) of any Manhattan distance observed. First, we observed the p-values of TPCA signature for protein pairs obtained with 10 compared to 3 or 4 optimal temperatures identified are not significantly different (differences in p-values i.e. Δp mostly near 0, see figures below) and are very well correlated (average Pearson's and Spearman R of 0.90 and 0.95 respectively in the intact cell data, and 0.92 and 0.96 respectively in the cell lysate data) reflecting directionality is preserved with overall only slight variation in p-values (see figures below). Nevertheless, we define the set of PPI with very strong TPCA signal under 10 temperatures (defined as those with p-value < 0.01) as true positives (which can be viewed as the subset of known PPI with potentially high recurring rate), and its subset that do not possess TPCA signal (defined as those with p-value > 0.1) under 3 or 4 temperature as false negatives. Conversely, we define false positives as PPI with non-existent TPCA signal under 10 temperatures (p-value > 0.1) but exhibit strong TPCA signal under 3 or 4 temperatures (p-value < 0.01).

Legend: Difference in p-values computed for protein pairs using optimal subset of temperatures identified, and the full set of temperatures. (a-b) The symmetric density plots illustrates that the TPCA signature p-values of most protein pairs did not change significantly in intact cells and cell lysates respectively between the full temperature set and identified subset.

Legend: Scatter plot of TPCA signatures (p-values) for random protein pairs. Correlation of TPCA signatures (p-values) of protein pairs from 10 temperature points and identified subset. We observed the TPCA signatures (p-values) of protein pairs obtained with 10 and 3 or 4 temperatures are not significantly different but are well correlated (average Pearson's and Spearman R of 0.90 (a) and 0.95 (b) respectively in the intact cell data and 0.92 (c) and 0.96 (d) respectively in the cell lysate data).

In summary, we define false negative pairs as:

$$(pvalue_{10 \text{ temp}} < 0.01) \text{ AND } [(pvalue_{3 \text{ temp}} > 0.1) \text{ OR } (pvalue_{4 \text{ temp}} > 0.1)]$$

And false positive pairs as:

$$(pvalue_{10 \text{ temp}} > 0.1) \text{ AND } [(pvalue_{3 \text{ temp}} < 0.01) \text{ OR } (pvalue_{4 \text{ temp}} < 0.01)]$$

From the intact cell data, 582 PPI have very strong TPCA signal (p-value < 0.01) under 10 temperatures but 2 of them (0.34%) are without TPCA signal (p-value > 0.1) under 3 or 4 temperatures. A total of 3132 PPI have no TPCA signal (p-value > 0.1) under 10 temperatures, and only 7 (0.22%) of them exhibit TPCA signal under 3 or 4 temperatures (p-value < 0.01). For the cell lysate data, there are 607 PPI with TPCA signal, of which none (0%) are found to be false-negative under 3 or 4 temperature points. Of the 3675 PPI without TPCA signal under 10 temperatures, 22 protein pairs (0.60%) exhibit TPCA signal under 3 or 4 temperatures. Thus, using the 3 or 4 temperatures identified do not affect overall analysis of PPI with very few false positives and false negatives.

Intuitively, Slim-TPC, with less data points (temperatures), will involve some trade-offs although it facilitates more flexible experimental design and reduce variations from multiple MS run (batch effect). However, after comparing the results of protein pairs and complexes obtained by the traditional method with those obtained by Slim-TPCA analysis, we believe that the impact of these trade-offs is relatively small and acceptable, particularly

when considering the elimination of batch effect arising from multiple MS runs. For the modulated protein complexes, the original analysis was performed comparing data obtained from multiple batches of TMT experiments which undoubtedly increased stochastic variation and hence potentially more false positives (modulated protein complexes). Slim-TPCA, with reduced number of temperatures, permits multiple conditions to be analysed concurrently on MS, thus minimize stochastic variation between different batches of MS run. Nevertheless, we recommend that the community conduct an assessment before selecting a method and to determine whether Slim-TPCA or traditional TPCA is an appropriate method for the research topic which we had included in our main text (reproduced below).

“For other species, we recommend to first perform conventional TPCA experiments to determine the melting temperature and to verify the feasibility of the Slim-TPCA method in this species sample”

Comment 10 The authors use Pearson correlations to show that the values remain high from 10 temperatures to 3 temperatures but it is unclear whether this is an accurate representation of correlation considering that these data sets are most likely not normally distributed.

Response: Thank you for your feedback. We concur that the Pearson’s correlation coefficient may not be the most appropriate measurement if the data is not normally distributed. We recalculated the correlation of the distances of the protein pairs using the Spearman’s correlation coefficient and obtained same combination of temperatures and similar conclusions, i.e. the distances of the protein pairs can maintain high consistency with the distances calculated at ten temperature points when using three temperature points (37°C, 49°C, 58°C) and four temperature points (37°C, 46°C, 55°C, 61°C). Figures 2a and 2b, redrawn using the Spearman’s correlation coefficient, are shown below and have been replaced in the manuscript. We had elected to use Pearson’s correlation in earlier analysis based on the reasoning that the calculation of TPCA signature is based on absolute difference in value. Regardless, both Pearson’s and Spearman’s correlation coefficients obtained identical outcome.

(Fig. 2a and 2b, shown below)

Fig. 2. Selection of optimal combinations of temperature points for TPCA profiling. (a) Correlation of Manhattan distances for 100,000 random protein pairs calculated using 3 temperature points and 10 temperature points. The combinations of 3 temperature points $\in \{37^\circ\text{C}, T_a^\circ\text{C}, T_b^\circ\text{C}\}$, where $T_a, T_b \in \{40^\circ\text{C}, 43^\circ\text{C}, 46^\circ\text{C}, 49^\circ\text{C}, 52^\circ\text{C}, 55^\circ\text{C}, 58^\circ\text{C}, 61^\circ\text{C}, 64^\circ\text{C}\}$. (b) Correlation of Manhattan distances for 100,000 random protein pairs calculated using 4 temperature points with those calculated using 10 temperature points. The combinations of 4 temperature points $\in \{37^\circ\text{C}, T_a^\circ\text{C}, T_b^\circ\text{C}, T_c^\circ\text{C}\}$, where $T_a, T_b, T_c \in \{40^\circ\text{C}, 43^\circ\text{C}, 46^\circ\text{C}, 49^\circ\text{C}, 52^\circ\text{C}, 55^\circ\text{C}, 58^\circ\text{C}, 61^\circ\text{C}, 64^\circ\text{C}\}$.

Comment 11 The quality of data appears to be very good based on the results observed in the bioinformatic evaluation that is done throughout. At the same time, it is unclear whether the quality of the results from the new workflow would hold up in a discovery based experiment where annotations are not leveraged. Would novel protein relationships be detected in an output where there is now more noise (since 10 temperatures are no longer used)?

Response: Thank you for your feedback. In our humble opinion, it will be tough for TPCA methods by itself to be accurate enough for practical use for the discovery of novel protein interactions regardless of the number of temperatures. You could refer to our response to comment 3 of first reviewer for illustration on the conceptual difficulty of the task. Succinctly, the inherent challenge in using the approach to make de novo predictions is there are too many non-interacting protein pairs with very similar melting curves while interacting protein pairs can have significant but are not necessary most similarity in melting curves – as most interactions do not occur at maximum frequency (i.e. every protein is in a complex) such as for many signalling complexes. Nevertheless, based on prior knowledge of biology, Hashimoto et al. discovered and validated new virus-host protein interactions using the TPCA method, extending the utility of the TPCA method. Here, we performed a similar analysis of the novel virus-host protein interactions identified by Hashimoto et al. using the 3 temperature points and found that the subset of temperature points used still enabled the identification of the novel interaction. Specifically, the virus protein pUL52 was found to interact with the host proteins IFIT1 and IFIT2 using the TPCA method. Using the 3 temperatures identified, their interaction could still be detected ($p < 0.01$). This simulation proves that Slim-TPCA can still find novel PPI and demonstrates

the robustness of the proposed Slim-TPCA method.

Legend: Novel viral-host complexes identified in the previous study (*pUL36-TSC1-TSC2*, *pUL52-IFIT1-IFIT2*) could similarly be identified in the 3-temperature-points Slim-TPCA method with protein interaction information.

Comment 12 Figure 1c: While the analysis including whole cells and cell lysates is interesting, it does not align with the rest of this section. At this point in the narrative by the authors, the choice of distance measures is being chosen. It is unclear how the choice of sample preparation (i.e. lysis vs. whole cell) is being used to decide on the distance algorithm.

Response: Thank you for your feedback. Ideally, the distance algorithm identified should give the best result for both whole cells and lysates. Fig. 1C demonstrates that the Manhattan distance performs the best among several distance algorithms both in cell lysates and intact cells. Similarly, we also demonstrated that as the number of temperature points decreases, Manhattan distance also performs best, both in intact cells (Fig. 1d) and in cell lysates (Supplementary Fig.2, shown below).

(Supplementary Fig.2, shown below)

Supplementary Figure 2. Box plot of the ability of different measures in predicting PPI when used with fewer temperature points in the cell lysate data. All combinations of temperature were tested accordingly. Pearson's distances and Cosine distances are meaningless with 2 temperature points.

Comment 13 Figure 1d: Consider adding an axis break so that the values for the Cosine and Pearson can be seen. It should also be considered that the error bars for each are wide enough that they overlap with each other across pretty much all temperatures, including those at 10 temperatures. It is also unclear which bars at 10 temperatures correspond to the different methods as the colors are not clearly visible.

Response: Thank you for your feedback. We apologize not explaining Fig. 1d clearly which lead to misunderstandings.

Fig. 1d and Fig. S2 are box plots, so the horizontal bars on the figures indicate the interquartile ranges (IQRs) of the box plots instead. The box plots indicate the AUC of ROC values calculated for multiple sets of n temperature points. For example, for combination of 2 out of 9 temperatures, there will be 36 unique sets ($9!/(9-n)!n!$) where $n = 2$. Thus, boxplots were used to show the distribution of multiple AUC obtained. We apologize for not explaining the meaning of the box plot clearly in the legend. Pearson's correlation and the Cosine distance could not be computed with 2 temperature points, thus were not reflected on the figure. We have added a legend to clarify this. Thank you for the feedback on the bars for 10 temperatures. We had thickened the bars to improve visibility. Accordingly, we have modified the legend for Fig. 1d and Fig. S2. with that for Fig. 1d reproduced below:

“Box plot of the ability of different measures in predicting PPI when used with fewer temperature points in the intact cell data. Each box plot represents the distribution of AUC obtained for all unique combination sets for n out of 10 temperature points using different distance measurements. Pearson distances and Cosine distances cannot be computed with 2 temperature points.”

Comment 14 Figure 2: It is not clear what the criteria was for selecting 3 or 4 temperatures as acceptable for approximating the full 10 temperature experiment. It is also not clear what the trade-off (impact to false positives) is by only using 3 or 4 temperatures. As was stated previously, there is a decrease in AUC that appears to be minimal but there is also a large impact to error bars around each median AUC.

Response: Thank you for your feedback. In this work, we had derived a mathematical framework to quantify the similarity (or difference) in information obtained with 10 temperatures and its subsets. TPCA quantifies difference in solubility among proteins across different temperatures and conditions to analyze protein-protein interaction. Thus, we qualified similarity in such information for protein pairs to identify the subset of temperatures that maximally encapsulated information obtained from the full temperature set. Specifically, we identified specific subsets of 3 and 4 temperatures which generated information (distance between two proteins) that correlate highly to the information obtained with the full temperature set with Pearson's R of 0.93 and 0.96 respectively. In the revised manuscript, we had reanalyzed the data with Spearman's correlation obtaining values of 0.93 and 0.96 for the same 3- and 4-temperature subsets. This correlation analysis permits a quantitative assessment of information maintain (or lost) and we deemed a correlation of >0.9 is good criteria or trade-off.

In Fig 1d, we computed the AUC of ROC values for multiple sets of n temperature points. For example, for combination of 2 temperatures, there will be 36 unique sets. Boxplots were used to show the distribution of multiple AUC obtained.

Comment 15 Figure 2d. The definition of the Dynamic p-value is not clear based on what is described in the methods section. Is this purely a p-value calculated in a temporal experiment? It is unclear how this is calculated or how it compares/differs from 2c. Related publications (i.e. Tan CSH, Go KD, Bisteau X, Dai L, Yong CH, Prabhu N, Ozturk MB, Lim YT, Sreekumar L, Lengqvist J, Tergaonkar V, Kaldis P, Sobota RM, Nordlund P. Thermal proximity coaggregation for system-wide profiling of protein complex dynamics in cells. *Science*. 2018 Mar 9;359(6380):1170-1177. doi: 10.1126/science.aan0346. Epub 2018 Feb 8. PMID: 29439025.) was also consulted and it was not defined here.

Response: Thank you for your feedback. We have inconsistently used "TPCA dynamic p-value" and "TPCA modulation signature" in the manuscript, both of which express the same meaning and are criteria for detecting dynamic changes in complexes. In brief, it is statistical evaluation of difference observed in TPCA signature of a protein complex across two conditions. We apologize for any confusion created. For the sake of consistency, we have used the "TPCA modulation signature" throughout the revised manuscript.

Comment 16 Figure 2c through 2e: The term "Signature" is used to describe the values in these plots. Is the "Signature" different than the Euclidean distance? Is it the p-value for the Euclidean

distance? If not, it would be beneficial and more clear to use the same terminology throughout the manuscript to avoid confusion. The term "signature" is a bit qualitative to this reviewer and does not come across as a quantitative evaluation of melt.

Response: Thank you for highlighting the potential confusion. In this manuscript, “TPCA signature” is used as a convenient and qualitative term to describe “shows sign of thermally-induced proximity co-aggregation of proteins that possibly lead to clustering of data points” while Euclidean distance is one of the many measurements that can be used to quantify this signal. We had now defined the term on page 3 in the manuscript as “TPCA signature is used qualitatively to refer clustering of data points that possibly arise from thermally-induced proximity co-aggregation of proteins.”

Comment 17 Figure 2e. The meaning of this figure is not clear. It could be helpful to have more labeling of axes. What values are depicted by red vs blue; are these z-scores? It is not clear how the conclusion is made that 3 and 4 temperature results are comparable with the panel on the left of 2e. Is the similarity between the three concluded based on the fact that changes in colors are similar between the three sub-panels? There may be other ways to visualize the data so that it can be reinforced that 3 and 4 temperatures are comparable with the 10 temperature conclusions. It is also not clear what is meant by "matched as close as possible".

Response: Thank you for your feedback and suggestion on possible better ways to visualize the data. In Fig. 2e, we used the z-score of the dynamic protein complex, and we apologize for not detailing the algorithm for this score. We had now added a description of the z-score algorithm to the methods section.

“**Empirical statistical assessment of TPCA Modulation Z-score.** In addition to using the TPCA modulation signature to identify complexes with significant dynamic changes, we also borrow the TPCA modulation z-score from previous paper to detect complex dynamics after virus infection. We used $M_{X_{c,i}}$, where $M_{X_{c,i}} = \frac{1}{1+M_{c,i}}$ and $M_{c,i}$ is the average Manhattan distance of proteins in complex © at specific infection time point (i; 24, 48, 72, and 96 hpi or mock), calculated from 10, 4, or 3 temperature points. Z-scores are then calculated based on the null-distribution of $M_{X_{c,i}}$ generated from random protein complexes with same size of proteins as in comp© (c) with 10,000 iterations.”

In Fig. 2e, we wanted to illustrate that the TPCA modulation signature of the complexes obtained using Slim-TPCA are consistent with those obtained with conventional 10-temperature TPCA using the same visualization strategy (heatmap) adopted in the original work c. The heatmap has the advantages for being more compact where data for specific complexes can be presented. Nevertheless, we concur that other visualization strategy could be used, particularly for global comparison of data, thus we created scatter plots of the z-scores as Supplementary Fig.5.

(Supplementary Fig.5, shown below)

Supplementary Figure 5. Scatter plot of TPCA modulation z-scores. (a-d) The TPCA modulation z-scores obtained using three temperature points maintained a good correlation with those obtained by the conventional TPCA method. The four subplots correspond to complex TPCA modulation signatures at 24 h, 48 h, 72 h and 96 h post-infection, respectively.

Comment 18 Figure 3e. Figure 4d and 4e: The y-axis indicates that the values are percentages. Is this a typo? If these are actually percentages (and not fractions), the values seem to be extremely low.

Response: Thank you for your feedback. These are indeed typos in our manuscript. We have rectified the errors, and below is one of the modified figures (Fig. 3e)

Figure 3e. Number and functional class of convergent protein complexes identified with different algorithms. Compared to the absolute distance algorithm, the relative distance

algorithm identifies more dynamic complexes associated with cell cycle, according to the annotation in the CORUM database.

Comment 19 Figure 3f and 3g. It is difficult to tell which curves correspond to MTX treatment and which are DMSO. The labels on 3g are also missing.

Response: Thank you for your feedback. Accordingly, we have modified Fig. 3f and 3g, shown as blow:

Figure 3f-g. Curves of subunits in Complex 2792 under DMSO and MTX, with 10 temperature points and 4 temperature points, respectively.

Comment 20 Supplementary Figure S4: It is not clear what the difference is between the three tables listed.

Response: We apologize for the overlook. First, the original Supplementary Figure 4 becomes the current Supplementary Figure 7 after modifying the order. Randomly generating 500 complexes and using a distribution fit for their distances can be a good alternative to the conventional TPCA method of generating 100,000 complexes at random and greatly reducing the time required for the calculation. Supplementary Figure 7 presents the results of the average distance approximation of the complexes using the Beta distribution. The three tables listed values approximated by the Beta distribution for average distance, change in absolute distance, and change in relative distance, respectively. We have modified the legend of Supplementary Figure 7 accordingly, as shown blow:

“The results of using 500 complexes with different distances and various distributions to approximate the distances of 100,000 complexes. The three subplots correspond to parameters that use distributions to fit three different distances, namely average distance, change in absolute distance, and change in relative distance, respectively. The Beta distribution performs well in all three approximations of distance.”

Comment 21 Figure 4g and 4h: Are the complexes reported by the Beta method false positives? While it is initially encouraging that the method with lower computational expense reports all

complexes from the sample method, it would be good to know how many are false positives.

Response: Without the capability and expertise to replicate the virus infection experiment, we can only perform literature survey of the complexes identified with Beta method only. The beta distribution with absolute distance algorithm identified additional 19 more modulated complexes (Fig. 4G and Supplementary table 10-12). Overall, these complexes can be broadly classified into those involved in protein trafficking, RNA degradation, and epigenetic regulation through histone methylation. A portion of these complexes had been implicated in viral infection or virus replication. They include the Exocyst complex regulating vesicular trafficking (Zaman et al., 2021), the RNase MRP complex and mRNA decay complex involved in viral RNA degradation (Jaag et al., 2011 and Balistreri et al., 2014) as well as **NF- κ B** complex, a known hallmark of viral infection (Santoro et al., 2003). Many of the remaining complexes are involved epigenetic regulation through methylation of histone, a process known to be modulated during viral infection (Tsai et al., 2020). Many of these newly identified complexes just miss the statistical threshold with the bootstrapping algorithm. Thus, while we cannot exclude the Beta method could identify more false positives, many relevant ones missed by the original bootstrapping algorithm are also identified. It will be beyond the scope of this work to precisely determine the false positive rate but at least a portion of the new complexes have high possibility or had already been reported to be involved in viral infection.

Comment 22 Figure 5: In plots like this and those in the supplementary section, it would be beneficial if error bars were present on each of the points so that the variability from the replicate experiments can be evaluated. Error bars would allow the user to evaluate whether the conditions that are assumed from changes in curve shape are actually within the "noise" of the system.

Response: Thank you for your feedback. We agree with you that it is necessary to add error bars, so we made the figures with error bars added. However, for the sake of readability, we still use the figures without error bars in Fig. 5d, and the figures with error bars are placed in the supplementary figures. Below is a snapshot of the figures with error bars while remaining ones can be found in the Supplementary Figure S8-S15.

(Part of Supplementary Fig.11, shown below)

Supplementary Figure 11. Identified divergent complexes after 8 hours of glucose supplementation (TPCA modulation $p > 0.95$).

Comment 23 Supplementary Fig.14 and Supplementary Fig.15 are referenced in the text but could not be located in the supplementary material.

Response: We apologize for overlooking this. In the revised manuscript, the original Supplementary Figure 14 and Figure 15 become the current Supplementary Figure 17 and Figure 18 after modifying the order, and included in this revision (see **support information**).

Comment 24 The conclusion that the modified workflow has acceptable results when compared to the 10 temperature workflow is based on use of Pearson correlation. It is not clear that this is an acceptable measurement due to the fact that the data sets being compared are not normally distributed but are skewed.

Response: Thank you for your feedback. As mentioned before, we strongly agree with you that the Pearson correlation coefficient should satisfy the normal distribution of the data set, and it was an oversight for us not to consider this. We recalculated the correlation of the distances of the protein pairs using the Spearman correlation coefficient and obtained similar conclusions, i.e., Using three temperature points (37°C, 49°C, 58°C) and four temperature points (37°C, 46°C, 55°C, 61°C), the distances of the protein pairs maintain highest consistency with the distances calculated at ten temperature points. Figures 2a and 2b, redrawn using the Spearman's correlation coefficient, are shown below and have been replaced in the manuscript.

(Fig. 2a and 2b, shown below)

Fig. 2. Selection of optimal combinations of temperature points for TPCA profiling. (a) Correlation of Manhattan distances for 100,000 random protein pairs calculated using 3 temperature points and 10 temperature points. The combinations of 3 temperature points $\in \{37^\circ\text{C}, T_a^\circ\text{C}, T_b^\circ\text{C}\}$, where $T_a, T_b \in \{40^\circ\text{C}, 43^\circ\text{C}, 46^\circ\text{C}, 49^\circ\text{C}, 52^\circ\text{C}, 55^\circ\text{C}, 58^\circ\text{C}, 61^\circ\text{C}, 64^\circ\text{C}\}$. (b) Correlation of Manhattan distances for 100,000 random protein pairs calculated using 4 temperature points with those calculated using 10 temperature points. The combinations of 4 temperature points $\in \{37^\circ\text{C}, T_a^\circ\text{C}, T_b^\circ\text{C}, T_c^\circ\text{C}\}$, where $T_a, T_b, T_c \in \{40^\circ\text{C}, 43^\circ\text{C}, 46^\circ\text{C}, 49^\circ\text{C}, 52^\circ\text{C}, 55^\circ\text{C}, 58^\circ\text{C}, 61^\circ\text{C}, 64^\circ\text{C}\}$.

Comment 25 Assessment of some data could not be done due to some axes and legends not being fully complete.

Response: Thank you for your feedback. We apologize for the typo in the manuscript. We have gone through and modified these figures with axes and legends.

Comment 26 Since the Manhattan distance is chosen over the Euclidean distance, it might be beneficial for the authors to speculate why it is in principle a better approximation and whether this only applies for 3-4 temperature data sets (and not the 10 temperature set).

Response: Without mathematical proofs, we could only speculate why Manhattan distance is better than Euclidean distance. Manhattan distance is computed as the sum of differences while in Euclidean distance, differences are first squared followed by summation and square root. For Euclidean distance, big differences are amplified to much greater extent than small difference (non-linear). We guess the difference in TPCA scales linearly with frequency (or occurrence rate) of interactions that is better modelled by Manhattan distance. Based on Fig. 1b and 1d, Manhattan distance is slightly but consistently better for both lysate and whole cell data as well as for 3-4 temperature set and 10 temperature set.

Comment 27 Figure 1b: This figure is not currently referenced in the main text. It is unclear how the error bars on the AUC for 2 temperatures are smaller in magnitude than the other higher temperatures in the figure. It is expected that with a smaller number of data points, there would be a larger amount of variability around the mean. Perhaps consider including the standard deviation as a variable when determining the optimal number of temperatures. This is relevant since it is being argued that there is a large decrease in AUC with the lower number of temperatures. It is also important to note that while the magnitude in difference between the high and low number of temperatures is only 0.06 (0.71 and 0.65), this is a large number when considering analysis of proteomic data sets where number of proteins can approach 10,000. This is also relevant in Supplementary Figure 1 where described improvements in AUC between "Cityblock" and "Euclidean" distance measurements are on the order of 0.005.

Response: Thank you for your response. We apologize again for not explaining clearly the box plots in Figure 1. We have revised the content to correct the problem of not mentioning Fig. 1b in the content.

As mentioned in previous response, Fig. 1b is a box plot, so the horizontal bars on the figures indicate the interquartile ranges (IQRs) of the box plots. The box plots indicate the AUC of ROC values calculated for multiple sets of n temperature points. For example, for combination of 2 out of 9 temperatures, there will be 36 unique sets. For combination of 3 out of 9 temperatures, there will be 84 unique sets. Thus, boxplots were used to show the distribution of multiple AUC obtained. With more unique sets, the probability of outliers increases and hence the observed larger "error bar" which reflects distribution of AUC values. Essentially, analysis presented in Fig. 1b reveals the degree of compromise in prediction performance which in our opinion is marginal and acceptable considering the advantages offered with using less temperatures. For the identification of modulated protein complexes, which we advocate the use of TPCA for, using less temperatures permit samples from multiple experiment conditions and replicates to be analysed at the same time by MS with TMT reagents. This is important to eliminate batch variation in machine measurement and ensure the same peptides are quantified across samples to reduce false positives and facilitate downstream analysis. We had previously applied the best-performing combination of temperatures identified in one dataset to other datasets but observed inconsistency in performance likely to **idiosyncratic** effect.

We concur with your comment that the difference in AUC can be small but with substantial impact when the number of proteins is large, but when considering the use of TPCA to identify differentially modulated protein complexes, we felt the advantages offered by Slim-TPCA outweigh this consideration as number of protein complexes is currently manageable and false positives from batch effects are minimized. Presently, we do not advocate the use for TPCA for predicting novel PPI, at least not by itself. This is because the number of non-interacting protein pairs greatly outnumber interacting protein pairs. Based on your feedback, we had added the text below in the manuscript.

“Nevertheless, we note that while the decrease in AUC is marginal with fewer temperature

points, the difference can still be substantial in the prediction of protein-protein interaction when considering large number of proteins and possible protein pairs. Here, we had used PPI prediction to evaluate information content embedded in the full temperature set and its subsets. For the identification of modulated protein complexes, which we advocate the use of TPCA for, using less temperatures permit samples from multiple experiment conditions and replicates to be analysed concurrently on MS instrument using isobaric labelling reagents e.g. TMT. This is important to eliminate batch variation in machine measurement and ensure the same peptides are quantified across samples to reduce false positives and facilitate downstream analysis.”

Comment 28 Figure 1d: It is concluded in this first section that the Manhattan analysis is better than the other methods based on the higher median value. It is not discussed, however, that the error bars for each of these data sets are extremely large and that there are some methods that actually have smaller error bars (i.e. Chebyshev). Consider explaining what criteria were used for selecting Manhattan as compared to Euclidean. It may also be prudent in the discussion section to explain why the chosen method better approximates (based on the details of the calculation).

Response: Thank you for your response. We apologize again for not explaining clearly the box plots in Figure 1. As discussed previously, Fig. 1d as well as Supplementary Fig.2 are boxplots, so the horizontal bars on the figures indicate not the error bars but the interquartile ranges (IQRs) of the box plots.

In terms of the criteria used for selecting Manhattan distance over Euclidean distance, Fig. 1c and 1d illustrate this together. In Figure 1c, the Manhattan distance produces higher AUC values than the Euclidean distance with conventional TPCA algorithm using 10 temperatures, in both cell lysates and intact cells. In Fig. 1d, the median AUC of the temperature point combinations decreases with the number of temperature points selected, but the AUC of the Manhattan distance is consistently higher than that of the Euclidean distance, and still maintains a relatively high AUC at three temperature points if an appropriate combination of temperature points is selected.

We have also included our speculation in discussion why Manhattan distance is better approximate/metric for TPCA signal as below. We can only speculate as we lack formal proof.

“We speculate frequency or occurrence rate of protein-protein interaction likely scale linearly with TPCA signal, hence is modelled better with Manhattan distance compared with Euclidean and other distance measurements that amplify difference non-linearly.”

Comment 29 Figure 2c and 2d: The data for each axis does not appear to be normally distributed but instead appears to be skewed. Is the Pearson coefficient the correct value to

describe the correlation between the two axes or would something like the Spearman be more suitable for the non-normally distributed data? This is particularly important due to the fact that a large amount of the data skews at the higher values (presumably those with more significant p-values). For example, there are many points that would be below a cut-off of 2 on one axis but well above 2 for the other axis. See Schober, P., Boer, C. & Schwarte, L. (2018). Correlation Coefficients: Appropriate Use and Interpretation. *Anesthesia & Analgesia*, 126 (5), 1763-1768. doi: 10.1213/ANE.0000000000002864.

Response: Thank you for your feedback and suggested reference. As mentioned before, we strongly agree with you that the Pearson's correlation coefficient should satisfy the normal distribution of the data set, and it was an oversight for us not to consider this. We had initially elected to use Pearson's correlation coefficient to evaluate the similarity in information obtained as the final TPCA signature (and difference) is based on absolute difference (Manhattan distance) and not ranking. Nevertheless, we recalculated the correlations in Fig. 2c and 2d using the Spearman's correlation coefficient and obtained similar conclusions that both TPCA signatures and TPCA modulation signatures remain high consistency in Slim-TPCA method.

Figure 2. (c) Statistical significance of TPCA signature for CORUM complexes (p-value) quantified with 3 and 4 temperature points as compared to 10 temperature points. (d) Statistical significance of dynamic (modulated) TPCA signatures for CORUM complexes (TPCA Modulation Signature) quantified with 3 and 4 temperature points as compared to 10 temperature points.

Comment 30 Figure 3a and 3b. The use of a Pearson correlation is most likely not suitable due to the fact that both axes are not normally distributed.

Response: Thank you for your feedback. We also redraw Fig. 3a and 3b using Spearman's correlation rather than Pearson's correlation and obtained similar conclusions that the TPCA modulation signatures obtained using relative distance are in good consistency with those obtained using absolute distance, both in the 10-temperature TPCA and the 4-temperature Slim-TPCA.

Fig. 3. *Relative distance algorithm identifies more dynamic complexes in TPCA and Slim-TPCA. (a-b) Correlation of the TPCA Modulation Signature of CORUM complexes obtained using the relative distance algorithm and the absolute distance algorithm for 10 and 4 temperature points, respectively.*

Comment 31 In the second paragraph of the last section prior to discussion, the following statement is made: “Three sets of biological replicates are performed, and the data collected are pooled for integrative analysis to maximize signal and reduce stochastic variation.” It is not clear what this statement means and whether the variability observed from the replicates was truly utilized in the analysis.

Response: In our algorithm, we computed the average value across replicates for each protein which should be nearer to the true value conceptually. As not all proteins are identified across replicates, average values are computed only proteins identified across 2 or more replicates are used subsequently to calculate Manhattan distance. Specifically, weighted averages based on number of PSMs are computed. We adopted this data processing strategy to maximize the number of proteins so that there will be more protein complexes with data for at least 3 subunits for analysis. We have now added description of how data are combined in the main text. As the averages of proteins might be derived from 2 replicates only, we adopted statistical bootstrapping approach (sampling) to derive the global fluctuation of random protein sets (to mimic protein complex) which is then used to estimate statistical significance of variation observed for real complex. Hence, variations in readings of individual proteins are not used. Instead, variations among random protein sets are used to assess significance in variation of TPCA signature.

Comment 32 The authors use the K562 cell line for the identification of optimal temperatures for the simplified version of their workflow. The authors do use a data set collected using a fibroblast cell line as a comparator. It would be beneficial, however, for the optimal temperatures to be compared using a different species (i.e. yeast) where the mean melt temperature could be higher or lower than that of K562. This would not necessarily involve generation of a novel data set but could be done using a data mining approach. This is particularly important if this workflow is recommended for use beyond that of mammalian

systems alone.

Response: Thank you for your feedback. In the manuscript, we provide suggestions for the selection of temperature points in mammalian cells. Then, we suggest picking the intermediate, widely separated temperature points that are evenly distributed around average melting temperature (T_m) will maximally encapsulated TPCA signal embedded across temperature range tested. However, we did not test this hypothesis using proteomic data from other species. In the revision, we analysed the proteomics data of *Arabidopsis thaliana* (Volkening et al., 2019) as well as thermal profiling data of 15 other species housed in Meltome database (Jarzab et al., 2020) and obtained similar results (see figure below). In summary, spearman's correlation coefficients between 2 optimal denaturing temperatures and full temperature set are around 0.95 except for *O.antarctica*. We have included result of this analysis in the manuscript as shown below

“Importantly, the identified optimal combination of temperature points is in line with our expectation that picking the intermediate, widely separated temperature points that are evenly distributed around average melting temperature (T_m) will maximally encapsulated TPCA signal embedded across temperature range tested. To test this theory, we chose TPP/MS-CETSA data from *Arabidopsis thaliana*⁴⁰, whose plant proteome has a lower melting temperature than the mammalian proteome. Similarly, the combination of temperatures producing data that correlate most with that from 10 temperature data (Supplementary Fig.6) are also intermediate, widely, and equally distributed around the average melting temperature of the proteome of *Arabidopsis thaliana* (~46.6°C). Furthermore, we also analyzed datasets from 15 species housed in the Meltome database⁴¹ that include *B. subtilis* and *C. elegans* etc. The data for these species similarly demonstrate that the right combination of temperature points in the Slim-TPCA method correlate well with data derived from full temperature set. These results suggest the framework proposed in this work could also be adopted for other species. For other species, we recommend to first perform conventional TPCA experiments to determine the melting temperature and to verify the feasibility of the Slim-TPCA method in this species sample.”

(Supplementary Fig.6, shown below)

Supplementary Figure 6. Heatmap of correlation between Manhattan distances of 100,000 random protein pairs computed with 3 and 10 temperature points in different species. Data were obtained from studies on *A. thaliana* and from the Meltome database. These results suggest the framework proposed in this work could also be adopted for other species.. (a) *A. thaliana*. (b) *B. subtilis*. (c) *C. elegans*. (d) *D. melanogaster*. (e) *D. rerio*. (f) *E. coli_cells*. (g) *E. coli_lystate*. (h) *Jurkat*. (i) *K562*. (j) *M. musculus_BMDC*. (k) *M. musculus_liver*. (l) *O. antarctica*. (m) *P. torridus*. (n) *S. cerevisiae*. (o) *T. thermophilus_cells*. (p) *T. thermophilus_lystate*.

Comment 33 It would be beneficial for the authors to describe the sample heating procedure that was used as it is integral to the experiments described. It is also stated at the end of the cell culture experiment that a Bradford assay was conducted; was the protein concentration adjusted prior to heat treatment?

Response: In the revised manuscript, we add more details about how we did the heat treatment and rewrote the *Cell culture* part in the *Materials and Methods* part as follow:

“**Cell culture.** K562 cells were cultured either in RPMI 1640 medium (Gibco) or glucose-free RPMI 1640 medium (Gibco) for 0 h, 4 h, 8 h, 24 h, and 48 h supplemented with 10%

FBS (PAN) and 1% penicillin–streptomycin (Gibco) at 37 °C, and 5% CO₂ in a humidified environment. Cells were washed twice with ice-cold PBS prior resuspended at 150 ul PBS (Gibco). Each condition sample was distributed in parallel into three aliquots and subsequently heated in parallel in a PCR (VWR, Doppio Gradient) block for 3 min to the three temperatures (37 °C, 46 °C, 58 °C). Then, cell subjected to (2X) lysis buffer containing concentration of 100 mM HEPES pH 7.5, 20 mM MgCl₂, 10 mM β-Glycerophosphate (sodium salt hydrate), 2 mM Tris(2-carboxyethyl) phosphine hydrochloride (TCEP), 0.2 mM Sodium orthovanadate, 0.2%(w/v) n-dodecyl β-D-maltoside (DDM), and EDTA-free protease inhibitor ((Sigma-Aldrich, USA). Cell suspension was subjected to five times flash-freezing in liquid nitrogen and rapid thawing in water to facilitate cell lysis. After centrifugation at 21000 g for 20 min at 4 °C, the supernatant was transferred to a new tube and the protein concentration was measured by the BCA assay kit (Thermo Fisher Scientific, USA). Then, samples heated to 37°C were taken 10 ug of protein, and the same volume of protein was taken at other temperature points to prepare the MS samples.”

Comment 34 While the authors clearly state the benefits of using the modified workflow (i.e. time and sample throughput), they do not discuss the deficiencies when compared to the original 10 temperature method. Possibly another way to state would be whether the authors are arguing that the original method using Euclidean distance is no longer a valid approach for measuring protein association.

Response: Thank you for your feedback. Based on your earlier feedback (comment 9), we had analysed the trade-offs compared to the original method. We have now highlighted the potential deficiencies in discussion (reproduced below). The traditional TPCA method with 10 temperature points is not invalid, just that we are proposing a new optimized method particularly for reducing batch effects when comparing multiple cellular conditions or performing time series experiments. We recommend that the community can decide whether to use Slim-TPCA depending on their research.

“In summary, we have incorporated multiple improvements for TPCA analysis of protein complex dynamics that include new statistical model and using a subset of temperatures. The latter permits multiple conditions and replicates to be analysed concurrently on MS instrument using multiplexing reagents. This eliminates batch variation across multiple MS runs in the traditional TPCA analysis that contribute to reduce false positives. Nevertheless, the optimization using reduced number of temperatures could also involve trade-offs, for example, protein interactions that can be identified by traditional TPCA methods may be missed in Slim-TPCA, or vice versa, Slim-TPCA may identify false-positive protein interactions. Importantly, while we had quantitatively assessed the loss of information using subset of optimal temperatures to be marginal, the analysis is performed in the perspective of analysing dynamics of protein complexes with TPCA. Thus, the findings and conclusions presented are not necessarily transferrable to other uses of protein thermal solubility data. With users mindful of these, we envision Slim-TPCA and associated software package to expediate functional characterization of existing and newly identified protein complexes.”

Comment 35 The principle of drug-protein interaction affecting protein melt temperature is actually a bit more established than that suggested by the references listed by the authors. It would be helpful to list older references such as:

Lo, M. C., Aulabaugh, A., Jin, G., Cowling, R., Bard, J., Malamas, M., & Ellestad, G. (2004). Evaluation of fluorescence-based thermal shift assays for hit identification in drug discovery. *Anal Biochem*, 332(1), 153-159. doi:10.1016/j.ab.2004.04.031

Cimpmperman, P., Baranauskiene, L., Jachimoviciute, S., Jachno, J., Torresan, J., Michailoviene, V., . . . Matulis, D. (2008). A quantitative model of thermal stabilization and destabilization of proteins by ligands. *Biophys J*, 95(7), 3222-3231. doi:10.1529/biophysj.108.134973

Response: Always great to know pioneer works. In the revised manuscript, these two references are cited now in the appropriate place:

“TPP/MS-CETSA permits the measurement of protein-ligand binding with endogenous proteins^{16,17}, and is widely used in drug target deconvolution¹⁸ and off-target studies¹⁹ in intact cells.”

Comment 36 Unable to assess the following: Quality of the raw MS data. Quality of the Python based code. FACS gating strategy.

Response: The raw MS data can be requested from iProX with the dataset identifier PXD040078. Python code can be required online via <https://slim-tpca.readthedocs.io/en/latest/index.html>.

Legend: The gating strategy of the initial FSC/SSC gate ensured that most of the cell debris, bubbles, and laser noise interference (all in the FSC-low region) were excluded from the analysis area.

Reviewer #3:

Comment 37 The key assumption of TPCA method is that proteins of similar melting behaviors belong to the same complex, and the melting behavior shift of a protein complex member indicates the protein leaves or joins the complex. This is enabled by the relatively high number of experimental temperatures thus the high resolution of melting behaviors, on condition that the quantitative accuracy achieved in the mass spec-based proteomics assay is also high. However, protein-protein interaction is not the only factor that can affect protein thermal stability and a high resolution (aka, a high number of temperature and high quantitative accuracy) is needed in order to make convincing conclusions. In this manuscript, the authors reduced the number of temperatures to 3 or 4, compromising the resolution and the robustness of the previous well-established data normalization method. It is true that the assay efficiency is improved by reducing the number of temperatures, but the confidence in the conclusion is also compromised to some extent. The author did carry out a series of simulation based on the previous TPCA publication and another study to justify the reduced number of temperatures and selection of distance calculation method, but the transferability of the conclusion needs to be future demonstrated considering the potential over-fitting risk.

Response: We wholeheartedly concur that changes in protein thermal stability can be due to multiple factors where changes in PPI is one of them. The number of temperature points could certainly affect fitting of melting curve to infer T_m of proteins. However, in the TPCA method, the differences in protein solubility among proteins across temperatures/conditions are analyzed instead. Intuitively, the number of temperature points should affect analysis of protein-protein interaction, but it is unclear how drastic the change would change. Thus, in this work, we first performed a quantitative evaluation assessing how changes in temperature affect predictability of protein-protein interaction and observed that the decrease in predictability is overall marginal (Fig 1d). While this may be surprising, a mathematical analysis can be performed to explain the observation.

Specifically, we derive a mathematical framework to quantify the similarities in information obtained with 10 temperatures and its subsets. Ultimately, TPCA analysis quantifies difference in solubility among proteins across temperatures and conditions to analyze protein-protein interaction. Thus, we quantified similarity in such information for protein pairs to identify the subset of temperatures that maximally encapsulated information obtained from the full temperature set. We identified specific subsets of 3 and 4 temperatures which generated information that correlate highly to the information obtained with the full temperature set where Pearson's R is 0.93 and 0.96 respectively. In the revised manuscript, we reanalyze the data to obtain Spearman's $R = 0.93$ and 0.96 for the same 3- and 4-temperature subsets. Thus, information computed from optimal subset of 3 or 4 temperature are very similar to information computed from 10 temperatures.

Thus, the compromise in resolution for TPCA analysis using less temperatures compared to a full temperature set is quantitatively assessed. We consider the compromise to be acceptable considering that using less temperatures allows samples from multiple

conditions and replicates to be analyzed at the same time on MS instrument (with TMT reagents). This serves to minimize instrumental batch effect and hence reduce false positives in the identification of modulated protein complexes.

In the original TPCA analysis, the statistical significance (p-value) of difference in protein distance (difference in thermal solubility between protein pairs) observed across conditions are estimated through a bootstrapping algorithm. We also confirmed that conclusion is transferable also for this downstream analysis where we observed high correlation in p-values obtained with full set of temperature and that of identified temperature subsets. This is performed comparing data obtained from methotrexate-treated cells and DMSO-treated cells (control). Very encouragingly, we obtained Spearman's $R = 0.90$ and 0.93 and Pearson's $R = 0.92$ and $= 0.94$ for the optimal 3 and 4 temperature set identified separately from another dataset (Fig 2d and Supplementary Fig.3d).

(Fig 2d, shown below)

Fig.2d Statistical significance of dynamic (modulated) TPCA signatures for CORUM complexes (TPCA Modulation Signature) quantified with 3 and 4 temperature points as compared to 10 temperature points.

(Supplementary Fig.3d, shown below)

Supplementary Fig.3d Statistical significance of dynamic (modulated) TPCA signatures for CORUM complexes (TPCA Modulation Signature) quantified with 3 and 4 temperature points as compared to 10 temperature points.

Similarly, we are also concerned whether the identified temperature subsets and conclusion is transferrable or conserved to other datasets. To verify this, we performed analysis on virus infection data generated in other lab with the identified temperature subsets, checking whether modulated protein complexes identified previously by the authors could still be identified. In the work, Z-score is computed instead of p -value. To ensure maximum coherence with their conclusion, we computed the Z-score as described using full temperature set and our identified subsets and observed high overall Pearson's R of around 0.85 (Supplementary Fig.5, figures above) between them. In the work, a novel interaction between host and viral protein was inferred from analysis of data and experimentally verified. Specifically, the virus protein pUL52 was found to interact with the host proteins IFIT1 and IFIT2 from TPCA analysis. Our new Slim-TPCA using the identified temperature subset could still revealed the novel interaction with p -value < 0.01 . In the seminal work where the concept of TPCA was first proposed (Tan et al, 2018), a modulated complex identified by TPCA was experimentally validated. This complex is still identified as modulated with our streamlined TPCA workflow producing p -value < 0.05 .

On the general transferability of the conclusion, the identified subsets of temperature points are in line with our expectation that picking the intermediate, widely separated temperature points will maximally encapsulated TPCA signal. As suggested by other reviewer, similar conclusion is obtained from the analysis of thermal stability/solubility data from *Arabidopsis thaliana*. Similarly, the combination of temperatures producing data that correlate most with that from 10 temperature data (Supplementary Fig.6a) are also intermediate, widely, and equally distributed around the average melting temperature of the proteome of *Arabidopsis thaliana* ($\sim 46.6^\circ\text{C}$). Furthermore, we also analysed datasets from 15 species housed in the Meltome database that include *B. subtilis* and *C. elegans* etc. The data for these species similarly demonstrate that the right combination of temperature points in the Slim-TPCA method can represent the results of 10 temperature points well (Supplementary Fig.6b-6p). Overall, the Spearman's correlation coefficients between 2 optimal denaturing temperatures and full temperature set are around 0.95 except for *O. antarctica* which still have a maximum Spearman's correlation coefficients of 0.82. Thus, the analytical approach and conclusion presented is transferrable to other species also.

Supplementary Figure 6. Heatmap of correlation between Manhattan distances of 100,000 random protein pairs computed with 3 and 10 temperature points in different species. Data were obtained from studies on *A. thaliana* and from the Meltome database. These results suggest the framework proposed in this work could also be adopted for other species.. (a) *A. thaliana*. (b) *B. subtilis*. (c) *C. elegans*. (d) *D. melanogaster*. (e) *D. rerio*. (f) *E. coli_cells*. (g) *E. coli_lystate*. (h) *Jurkat*. (i) *K562*. (j) *M. musculus_BMDC*. (k) *M. musculus_liver*. (l) *O. antarctica*. (m) *P. torridus*. (n) *S. cerevisiae*. (o) *T. thermophilus_cells*. (p) *T. thermophilus_lystate*.

To further validate the utility of our streamlined TPCA workflow in study of biological processes, we had applied streamlined TPCA on K562 deprived of glucose. Samples are collected across multiple time points, subjected to heating at 3 temperatures and analyzed concurrently on MS using TMT16. Our analysis revealed most protein complexes had dissociated under glucose deprivation, thus conform to the expected down-regulation of most basal cellular activities. Here, in this revision, we performed Co-IP to experimentally validate two convergently modulated protein complexes identified (Figure 6a). One of them is the Emerin complex 1 which function to organize the nuclear membrane during cytokinesis, while the other is the USP22-SAGA complex which serves as a regulatory center for signaling, chromatin modification, DNA damage repair, and gene control.

Supplementary Figure 21. TPCA profiling to K562 cells under glucose deprivation. (a-b) TPCA profile of two protein complexes Emerin complex 1 and USP22-SAGA complex at different time point of glucose deprivation.

Our streamlined analysis suggests increased association of Emerin with MYH9 during glucose deprivation (Supplementary figure 21) which was recapitulated in our Co-IP experiments (Fig.6d-e, shown below also). In addition, modulation of USP22-SAGA complex (CORUM ID: 6641) was also validated (Fig. 6d-e). Specifically, data from streamlined TPCA workflow suggests dissociation of TAF9B and TADA3K which was recapitulated with Co-IP experiments. Results of these two Co-IP experiments are added in our revision in Figure 6a.

Fig. 6. (d-e) K562 cells were incubated in presence and absence of glucose for 48 h, and analyzed by Immunoprecipitation and immunoblotting using MYH9 antibody, Emerin antibody, TAF9B antibody and TADA3L antibody. GAPDH antibody was used as loading control.

We have added the following to the main text and *Materials and Methods* part:

“Main text: Finally, we performed Co-IP to experimentally validate two convergently modulated protein complexes identified (marked red in Figure 6a). One is Emerin complex 1 which function to organize the nuclear membrane during cytokinesis⁵⁷, and the other is USP22-SAGA complex which function is a regulatory center for signaling, chromatin modification, DNA damage repair, and gene control⁵⁸. The extent of TPCA signals of these two complexes at 24h and 48h of glucose deprivation was relatively consistent (Supplementary Fig.21). Slim-TPCA analysis revealed increased association between subunits for both of Emerin complex 1 and USP22-SAGA complex during glucose deprivation which was recapitulated in our Co-IP experiments (Fig. 6d). After 48 h of glucose deprivation, the TPCA signals of the proteins used for experimental validation converged gradually.”

“Materials and Methods: Preparation of Co-immunoprecipitation (Co-IP) samples. K562 cells under glucose or glucose-free conditions for 48 h were harvested. Cells were lysed using RIPA buffer as above, sonicated, and then centrifuged at 21000 g for 20 min at 4 °C. The supernatant was transferred to a new tube and the protein concentration was measured by the BCA assay. Total 1.5 mg cell lysates were then incubated with 4 µg of primary antibody diluted in 700 µl of PBST (Rabbit pAb Control IgG (AC005, Abclonal), TAF9B antibody (28713-1-AP, Proteintech) and Emerin antibody (10351-1-AP, Proteintech) in a rotation wheel overnight at 4°C. Next, samples were incubated with Protein A/G Magnetic Beads (HY-K0202, MCE) for 6 h at 4°C. To remove the unbound antibody, beads were washed fifth times with 1 ml of PBS. 50 µl of protein loading buffer was added to the beads after which they were boiled at 95 °C for 10 mins. Samples were then loaded on an SDS-PAGE gel and further processed for western blotting.”

All in all, we had validated the robustness and transferability of streamlined TPCA and temperature subsets in other species/system (e.g. host-viral interaction, Arabidopsis thaliana, drug treatment) as well as to downstream analysis (bootstrapped p-value and z-score). We concur with the concerns of the reviewer, thus had performed extensive testing prior submission of the work. In the revised work, we further shown the conclusion is transferrable even to many other species, and also experimentally validated convergently modulated protein complexes identified with our new streamlined TPCA. In fact, we had already applied the streamlined TPCA with identified temperature subsets to study autophagy with collaborators with experimentally validated observations which we hope to share in a separate work soon.

We also agree wholeheartedly that quantitative accuracy is crucial. High number of

temperature points could mitigate errors and random fluctuation in MS quantification, such as for curve(model) fitting to smoothen out noise. Our own experience is errors and random fluctuation in MS quantification in bottom-up proteomics could also be mitigated (at least to some degree) by longer MS run quantifying more peptides per protein. However, a challenge when applying classic 1D-TPP protocol for time-series analysis of protein complex dynamics with TPCA is batch variation across MS runs. Variations in protein quantification are observed even between identical samples analysed in sequel that arise from semi-stochastic sampling of ions in DIA mode and inherent noise in instrument measurement. When performing time-series analysis of protein complexes, the need to ensure the same set of peptides are quantified is paramount to minimize erroneous conclusion arising from different proteoforms of proteins identified across different MS run. Identifying an optimal subset of temperatures that maximally encapsulated information embedded in the bigger set allow samples from multiple time points or conditions to be analysed concurrently by MS. This minimizes batch effort and maximizes the coherence of proteoforms identified. Importantly, identifying protein complexes spanning multiple time points with similar changes in TPCA signature (convergent or divergent) further increase the authenticity of observation and facilitate biological interpretation. Thus, streamlined TPCA with reduced temperature set allowing samples from multiple time point to be analysed concurrently with the same TMT can increase robustness.

We are grateful to the reviewer for insight and concern that we hope to convince the method presented can expediate biological discovery. Accordingly, based on feedback given, we had now included a more thorough discussion to better inform users and reader the pros and cons of the new streamlined TPCA method (reproduced below).

“In summary, we have incorporated multiple improvements for TPCA analysis of protein complex dynamics that include new statistical model and using a subset of temperatures. The latter permits multiple conditions and replicates to be analyzed concurrently on MS instrument using multiplexing reagents. This eliminates batch variation across multiple MS runs in the traditional TPCA analysis and reduce false positives. Nevertheless, the optimization using reduced number of temperatures could also involve trade-offs, for example, protein interactions that can be identified by traditional TPCA methods may be missed in Slim-TPCA, or vice versa, Slim-TPCA may identify false-positive protein interactions. Importantly, while we had quantitatively assessed the loss of information using subset of optimal temperatures to be marginal, the analysis is performed in the perspective of analyzing dynamics of protein complexes with TPCA. Thus, the findings and conclusions presented are not necessarily transferrable to other uses of protein thermal solubility data. With users mindful of these, we envision Slim-TPCA and associated software package to expediate functional characterization of existing and newly identified protein complexes.”

Comment 38 Besides, the measured protein melting behavior is also affected by other

experimental conditions, including lysis buffer composition, in cell vs lysate. Thus, the emphasis on the optimal three or four temperatures may mislead the community.

Response: Thank you for your feedback. We strongly agree with you that protein thermal profiling can be influenced by many factors and the ability of the TPCA method to detect PPI also varies with the temperature point used. As we mentioned before, in the revision, we analysed the trade-offs in detecting PPI brought by reducing the temperature point. At the same time, however, Slim-TPCA also offers some advantages, such as simplifying the experimental process and eliminating batch effects. We had added a more thorough discussion on the pros and cons including issues raised by the reviewer in the revised manuscript to be inform the community who may use Slim-TPCA in their research.

In addition, we are also aware that our suggested choice of three or four temperature points is based on data from the K562 cell line, but in samples from different experimental conditions or species, Slim-TPCA suitability and temperature point selection are likely to be different. Accordingly, we had performed analysis of TPP data obtained from 16 other species which reveal similar conclusion that optimal subsets of temperatures can encapsulated most of the information embedded within 10 temperatures for the purpose of TPCA analysis. Briefly, the spearman's correlation coefficients between 2 optimal denaturing temperatures and full temperature set are around 0.95 except for *O. antarctica* (Supplementary Fig.6, and figures in comment 37). Nevertheless, this does not imply the conclusion is transferrable to other use of TPP data which we had added in our discussion. Accordingly, we have added text below in our discussion.

“Importantly, while we had quantitatively assessed the loss of information using subset of optimal temperatures to be marginal, the analysis is performed in the perspective of analysing dynamics of protein complexes with TPCA. Thus, the findings and conclusions presented are not necessarily transferrable to other uses of protein thermal solubility data.”

Comment 39 Considering the limited number of temperatures used in the showcase experiment (glucose deprivation assay), the conclusion more of demonstrated the protein thermal stability changes after glucose deprivation. However, the thermal stability changes may not necessarily indicate protein-protein interaction changes, it could be PTM changes, protein localization changes, etc. It is a stretch to say “Proteome-wide Protein Complex Dynamics” in this case. It may work for some essential, large and abundant protein complexes, but the robustness for proteome-wide protein complexes is still unclear.

Response: While we agreed that changes observed in data obtained could arise from many possible factors, we had been careful restricting our analysis only to well-annotated protein complexes. In fact, the glucose deprivation experiment had been carefully picked to evaluate the “proteome-wide” coverage of the streamlined TPCA, as we will expect down-regulation of most basal activities during glucose deprivation, and as such the dissociation of more protein complexes compared to other conditions. Indeed, we observed about twice as many protein complexes on average that displayed divergent TPCA signature (complex

dissociation) than protein complexes with convergent TPCA signature. In comparison, we observed more complexes with convergent than divergent TPCA signature for all other cellular conditions profiled with the streamlined TPCA method. The modulated complexes identified are of diverse functions including those implicated chromatin remodelling, transcription, apoptosis, cell cycle, protein transport, protein degradation and signalling (Fig 6a and Fig S8-S15). Nonetheless, while we believe our experiment demonstrates “proteome-wide” coverage as modulated protein complexes involved in diverse functions were identified, proteins annotated in CORUM database still represent only a subset of the proteome. Accordingly, we have change “proteome-wide” to “system-wide” throughout the manuscript, the original term used in Tan et al (2018).

Comment 40 Some summary statistics should be included in Fig5, for example, how many proteins are quantified, cv, how may protein complexes are investigated in total.

Response: Thank you for your response. We apologize for not listing this necessary information, which has been added to figure 5 and the in the revision, shown as blow:

“Three sets of biological replicates are performed where 6476, 6627 and 6411 proteins were identified respectively with 8311 unique proteins identified altogether.”

“Average values across replicates are computed for a total of 5813 proteins that appear in at least 2 replicates and survived filtering criteria (see Materials & Method) and are used for downstream analysis.”

“A total of 783 complexes from the CORUM database were investigated.”

We also conduct a CV analysis, shown in the Supplementary Figure 19:

Supplementary Figure 19. Coefficient of variation of proteomic soluble fraction across 3 biological replicates. In a set of TMT16 experiments, 15 channels corresponded to different experimental conditions, i.e., combinations of 3 temperatures (37, 49, 58°C) and 5 cell states (0, 4, 8, 24, 48h after glucose deprivation). The coefficient of variation at 58°C is greater than that at 49°C due to overall lower protein abundance, but is well within acceptability.

Comment 41 Is protein level FDR also controlled? Method only says 1% PSM and peptide FDR.

Response: FDR control for protein and peptide is 1% at strict level and 5% at relaxed level.

Comment 42 The effect size in many plots (e.g. Fig 5d and some supplemental figures) is not big. Some seems to fall in the range of measuring variance of mass spec-based proteomics. Were the three replicates carried out completely independently (from cell culture to data acquisition)? Adding error bars or confidence intervals would help.

Response: Thank you for your feedback. We agree with you that it is necessary to add error bars, so we made the figures with error bars added. However, for the sake of readability, we still use the figures without error bars in Fig. 5d, and the figures with error bars are placed in the supplementary figures. Here is a snapshot of Fig. 5d with error bars, more detailed information can be found in the supplementary figure. The replicates were carried out independently on different days i.e. biological replicates.

(Part of Supplementary Fig.11, shown below)

Supplementary Figure 11. Identified divergent complexes after 8 hours of glucose supplementation (TPCA modulation $p > 0.95$).

Comment 43 Supplementary Figure 5, there is really no difference between the two distance calculation methods looking at the plots.

Response: Supplementary Figure 5 (now Supplementary Figure 8 in the revised manuscript, snapshot provided below) depicts changes in thermal solubility of proteins across multiple time points (0, 4, 8, 24, 48 hours) for convergently modulated protein

complexes identified at 4th hour of glucose deprivation (non-exclusive). Minimal difference between the plots indicate observed changes are consistent across different time points. To improve clarity, we had now labelled the time points on top of the plots (see below). Difference between onset of glucose deprivation and other time points may not be obvious visually due to relative distance algorithm used which can detect small changes in overall difference in protein thermal solubility among multiple subunits of a protein complex.

Legend: Snapshot of previous Supplementary Figure 5.

Legend: Snapshot of current Supplementary Figure 5 (now supplementary Figure 8) with time point labels.

Comment 44 The method section has a paragraph describing “Treatment with Pharmacological Inhibitors”. But no main text/data/figure is related to this part of the method.

Response: We apologize for the miss named the method. In the revised manuscript, we rewrite this part in the *Materials and Methods* part as follow:

“**Preparation of WB samples.** K562 cells under glucose or glucose-free conditions for 8 h, 24 h, 48 h, and 52 h were harvested. Cells were lysed using RIPA buffer containing a final concentration of 50 mM Tris-HCL pH 8.0, 150 mM NaCl, 1% Triton x-100, protease cocktail, and 1 mM PMSF facilitated by freeze-thawing five times using liquid nitrogen.

After centrifugation at 21000g for 20 min at 4 °C, the supernatant was transferred to a new tube and the protein concentration was measured by the BCA assay. Cell lysates were analyzed by western blotting.”

Comment 45 The main text refers to Supplemental Figure 13, 14 and 15, but the figures are not available in the submission.

Response: We apologize for the careless mistake. In the revised manuscript, the original Supplementary Figure 13,14 and 15 become the current Supplementary Figure 16, 17 and 18 after modifying the order. (see support information).

Comment 46 The experimental details for the glucose deprivation assay (Fig 6) are missing in the method section.

Response: The experimental details for Fig.6b was mentioned in *Detection of ATP levels* in the *Materials and Methods* part. In the revised manuscript, we rewrote the *Materials and Methods* for Fig.6c in the *Materials and Methods* part.

Materials and Methods for Fig.6b:

“**Detection of ATP levels.** ATP measurement was determined by CellTiter-Glo Luminescent Cell Viability Assay (Promega, USA). K562 cells under glucose or glucose-free conditions for 0 h,8 h, 24 h, and 48 h were seeded in a 96-well plate at 10 thousand cells per well. Cells were lysed using Cell Titer-Glo® reagent and mixed for 2 min on an orbital shaker. The plate was incubated for 10 min and analyzed by microplate reader (EnSpire), while cells number were measured by blood counting plates.”

Materials and Methods for Fig.6c:

“**Preparation of WB samples.** K562 cells under glucose or glucose-free conditions for 8 h, 24 h, 48 h, and 52 h were harvested. Cells were lysed using RIPA buffer containing a final concentration of 50 mM Tris-HCL pH 8.0, 150 mM NaCl, 1% Triton x-100, protease cocktail, and 1 mM PMSF facilitated by freeze-thawing five times using liquid nitrogen. After centrifugation at 21000g for 20 min at 4 °C, the supernatant was transferred to a new tube and the protein concentration was measured by the BCA assay. Cell lysates were analyzed by western blotting.”

Comment 47 Figure legends for Supplemental Figure 5-12 are missing. Difficult to figure out what each panel indicates. It’s better to label each panel indicating whether it’s a convergent or divergent case.

Response: Thank you for your response. We sincerely apologize for our negligence in adding the figure legends. We have added figure legends in the revision. The original supplemental Figure 5-12 became supplemental Figure 8-15 after revision. For example, the legend of supplemental figure 8 is: “**Identified convergent complexes after 4 hours of glucose supplementation (TPCA modulation $p < 0.05$)**”. We have also labelled the time

point on top of the figures (see comment 43).

REVIEWER COMMENTS

Reviewer #1 (Remarks to the Author):

The authors have addressed all of my comments, and I am happy to recommend the article for publication in its revised form.

Reviewer #2 (Remarks to the Author):

Key results

The authors describe a workflow for identifying potential protein interactions based on the shape of their respective melt profiles. This workflow is based on their previously reported TPCA with the following modifications. First, the proposed workflow uses a smaller number of temperatures in the heating gradient. Second, the analysis uses a Manhattan distance rather than a Euclidean distance to describe the protein melt curve shapes. Third, their analysis uses relative distance instead of absolute distance. Fourth, they have developed a p-value calculation workflow that is less time intensive by using a Beta distribution rather than a bootstrapping approach. Their new data analysis pipeline has been published as a publicly available python program and was validated using a novel temporal data set.

Significance

This work and modified workflow significantly increase the throughput of a TPCA experiment and consequently decrease the level of noise encountered when analyzing samples in multiple MS experiments. The benefit of the modified workflow is even more significant when considering experiments where multiple drug treatments and time points are incorporated. The Slim-TPCA approach has great potential to advance the field and improve the quality of findings from TPCA experiments.

Data and methodology

The quality of data appears to be very good based on the results observed in the bioinformatic evaluation that is done throughout. The authors (since the last review) have incorporated an evaluation of the approach using multiple cell lines (via a publicly available database).

Suggested improvements

The authors have incorporated improvements that I suggested in the previous draft/review.

Reviewer #3 (Remarks to the Author):

Please see the attached.

The authors conducted additional analysis and included an assay to address certain concerns. However, the newly added content was still not convincing. Besides, some comments from reviewers remained unaddressed.

Fundamentally, the authors overstated the significance and the robustness of the simplified TPCA assay. Notably, it's a widely recognized fact that 30-40% of the quantified proteins in a TMT-based thermal protein profiling assay don't fit well to sigmoidal curves (Nature Methods volume 17, pages 495–503, 2020, and several other datasets published by Dr. Savitski's and Dr. Kuster's groups). In all non-curve-based assays, including simplified TPCA assay, data interpretation can be complex for the 30-40% proteins whose melting behaviors don't follow sigmoidal curves. The assumption that calculation from simplified temperatures represents areas under melting curves may not hold true for these proteins.

Moreover, a statistically significant difference without additional effect size cutoff in the simplified TPCA assay doesn't necessarily mean a biologically meaningful thermal stability changes. To this point, there was a concern in the 1st reviewing process: *"The effect size in many plots (e.g. Fig 5d and some supplemental figures) is not big. Some seems to fall in the range of measuring variance of mass spec-based proteomics. Were the three replicates carried out completely independently (from cell culture to data acquisition)? Adding error bars or confidence intervals would help"*. The authors simply added error bars to the figures, without addressing how the replicates were carried out, and how/whether the error bars/cv justify the tiny effect sizes. Being statistically significant doesn't necessarily indicate biological significance. Many of the claimed "significant" events are essentially quantifying 10%-20% difference in protein abundance, which is usually beyond the quantitative power of proteomics. For example, a convergent protein complex at 8 h was highlighted in supplemental figure 10 as shown below. It's easy to tell by eye that the variance at 49C and 58C are high at 0 hr and there is essentially no enough power to claim biological difference.

The authors changed the term "proteome-wide" to "system-wide" to address the comment *"Considering the limited number of temperatures used in the showcase experiment (glucose deprivation assay), the conclusion more of demonstrated the protein thermal stability changes after glucose deprivation. However, the thermal stability changes may not necessarily indicate protein-protein interaction changes, it could be PTM changes, protein localization changes, etc. It is a stretch to say "Proteome-wide Protein Complex Dynamics" in this case. It may work for some essential, large and abundant protein complexes, but the robustness for proteome-wide protein complexes is still unclear."* The term "proteome-wide" and "system-wide" are used very often interchangeably in the field. The reviewer was commenting on the translatability from thermal stability changes to protein complex assembly/disassembly. Thermal stability changes don't necessarily mean protein complex

assembly/disassembly. It could be from the PTM or PPI (transient or loose interactions with other proteins not belonging to a protein complex) changes of the free form of a protein subunit.

Overall, the manuscript still lacks the strength to meet the standard of Nature Communications in my opinion.

Comment 1 Fundamentally, the authors overstated the significance and the robustness of the simplified TPCA assay. Notably, it's a widely recognized fact that 30-40% of the quantified proteins in a TMT-based thermal protein profiling assay don't fit well to sigmoidal curves (Nature Methods volume 17, pages 495–503, 2020, and several other datasets published by Dr. Savitski's and Dr. Kuster's groups). In all non-curve-based assays, including simplified TPCA assay, data interpretation can be complex for the 30-40% proteins whose melting behaviours don't follow sigmoidal curves. The assumption that calculation from simplified temperatures represents areas under melting curves may not hold true for these proteins.

Response: We thank the reviewer for highlighting this which prompted us to reanalyse one of the datasets generated in Tan et al., Science 359(6380):1170-1177 that was published in 2018. Specifically, we fitted the TPP data generated from intact K562 cells (Suppl. Table 7) with the sigmoidal curve function listed in Nature Protocol 10(10):1567-93 (2015). Among the 8260 proteins quantified, 508 (6.15%) proteins could not be fitted with sigmoidal curves (either failing to fit or has $R^2 < 0.8$). This number rose to 893 (10.8%) for $R^2 < 0.9$ or not fitting at all.

Importantly, of the 3530 unique proteins in the list that are annotated in CORUM database, 102 proteins (2.89% vs. 6.15% globally) failed to fit or have $R^2 < 0.8$ while 160 proteins (4.53% vs. 10.8% globally) failed to fit or have $R^2 < 0.9$. Thus, proteins annotated in the CORUM database are under-represented in proteins which thermal solubility do not fit well to sigmoidal curve behaviour. We repeated our analysis using data obtained with intact HEK293T cells (Suppl. Table 21, Tan et al., Science 2018), and observed only 1.21% (vs. 2.77% globally) of proteins in CORUM complexes could not be fitted with sigmoidal curve or have $R^2 < 0.8$. The number is 2.23% (vs. 4.58% globally) for proteins that could not fit to sigmoidal curve or have $R^2 < 0.9$. We also observed similar trend using data from Savitski et al., 346(6205):1255784.

While we do not think the underlying basics of TPCA for analysing dynamics of protein complexes relies on the assumption of sigmoidal curve behaviour and areas under melting curves, the problem of non-sigmoidal behaviour is minimal given the small number and under-representation of non-sigmoidal behaving proteins in CORUM protein complexes.

Comment 2 Moreover, a statistically significant difference without additional effect size cutoff in the simplified TPCA assay doesn't necessarily mean a biologically meaningful thermal stability changes. To this point, there was a concern in the 1st reviewing process: "The effect size in many plots (e.g. Fig 5d and some supplemental figures) is not big. Some seems to fall in the range of measuring variance of mass spec-based proteomics. Were the three replicates carried out completely independently (from cell culture to data acquisition)? Adding error bars or confidence intervals would help". The authors simply added error bars to the figures, without addressing how the

replicates were carried out, and how/whether the error bars/cv justify the tiny effect sizes. Being statistically significant doesn't necessarily indicate biological significance. Many of the claimed "significant" events are essentially quantifying 10%-20% difference in protein abundance, which is usually beyond the quantitative power of proteomics. For example, a convergent protein complex at 8 h was highlighted in supplemental figure 10 as shown below. It's easy to tell by eye that the variance at 49C and 58C are high at 0 hr and there is essentially no enough power to claim biological difference.

Response: We apologize that, while we had added the error bars to the figures as explicitly suggested, we overlooked comprehending the feedback further to comment on the error bars and the measuring variance. We thank the reviewer for this valuable feedback as we felt this should be highlighted to readers and potential users of Slim-TPCA on using error bars to guide interpretation of obtained data.

In fact, in our very first version of manuscript submitted, we had already devoted a section explaining the possible pitfall of using the relative distance scoring to identify modulated protein complexes. Accordingly, we added this suggestion by reviewer to this part of the manuscript (reproduced whole below with new addition in blue).

Conceptually, the use of relative distance to quantify changes in TPCA signature across conditions can reduce false negative to identify more modulated protein complexes but the approach can also lead to a higher false positive rate. The latter is particularly more pronounced if quality of the data is noisier, particularly when authentic biological difference is smaller than intrinsic instrumental noise. In this case, the use of error bars or standard deviation could guide interpreting authenticity of difference observed. With Slim-TPCA, multiple replicates can be analyzed simultaneously by MS instrument to at least minimize variance from batch measurement. We previously reported that precision or reproducibility in thermal solubility quantified depends in part on the number of peptide-spectral match (PSM) and ion intensity of proteins (Tan et al. Science 2018. To reduce the false positive rate, proteins with less precise values can be filtered based on these criteria. Alternatively, data from multiple technical and biological replicates can be integrated to obtain averages that are nearer to the true values. We prefer the latter approach as the former approach of filtering based on PSM and ion intensity can remove many proteins. Using data integrated from three biological replicates reported in previous work, we found the relative distance identified more modulated protein complexes that correlate with expected biological activities.

That said, for identified modulated USP22-SAGA complex (CORUM ID: 6641) which was validated with Co-IP experiments in our previous revision, we observed considerable variance in MS measurement for its TAF9B subunit at 48th hours where the Co-IP experiments were carried out. However, enhanced association between TAF9B and TADA3 was consistently observed across the three Co-IP biological replicates performed. Thus, authentic biological effect could also be masked as result based on error bars.

Regarding how replicates were carried out, we had stated in the rebuttal letter how they were obtained, which we had reproduced and highlighted below. In the main manuscript, we had also stated three biological replicates were collected as “*Three sets of biological replicates are performed where 6476, 6627 and 6411 proteins were identified respectively with 8311 unique proteins identified altogether.*”

Thus, a portion of high variance observed could arise from biological variation. While performing biological replicates on different days could increase biological variance in addition to instrument/measurement variance, we have elected to do so with the hope of improving biological reproducibility, which in our humble opinion should be a priority over technical/measurement consistency.

Comment 42 The effect size in many plots (e.g. Fig 5d and some supplemental figures) is no tbig. Some seems to fall in the range of measuring variance of mass spec-based proteomics. Were the three replicates carried out completely independently (from cell culture to data acquisition)? Adding error bars or confidence intervals would help. Response: Thank you for your feedback. We agree with you that it is necessary to add error bars, so we made the figures with error bars added. However, for the sake of readability, we still use the figures without error bars in Fig. 5d, and the figures with error bars are placed in the supplementary figures. Here is a snapshot of Fig. 5d with error bars, more detailed information can be found in the supplementary figure. **The replicates were carried out independently on different days i.e. biological replicates.**

Comment 3 The authors changed the term “proteome-wide” to “system-wide” to address the comment “Considering the limited number of temperatures used in the showcase experiment (glucose deprivation assay), the conclusion more of demonstrated the protein thermal stability changes after glucose deprivation. However, the thermal stability changes may not necessarily indicate protein-protein interaction changes, it could be PTM changes, protein localization changes, etc. It is a stretch to say “Proteome-wide Protein Complex Dynamics” in this case. It may work for some essential, large and abundant protein complexes, but the robustness for proteome-wide protein complexes is still unclear.” The term “proteome-wide” and “system-wide” are used very often interchangeably in the field. The reviewer was commenting on the translatability from thermal stability changes to protein complex assembly/disassembly. Thermal stability changes don’t necessarily mean protein complex assembly/disassembly. It could be from the PTM or PPI (transient or loose interactions with other proteins not belonging to a protein complex) changes of the free form of a protein subunit.

Response: We thank the reviewer for the elaboration which we now begin to appreciate. Accordingly, we had replaced the term “system-wide” in the manuscript with “at scale” or “*en masse*” whenever appropriate.